# CUARewardBench: A Benchmark for Evaluating Reward Models for Computer-Using Agents

**Haojia Lin** [* 1]  **Xiaoyu Tan** [* 1]  **Yulei Qin** [* 1]  **Zihan Xu** [1]  **Yuchen Shi** [1]  **Zongyi Li** [1]  **Gang Li** [1]  **Shaofei Cai** [3]
**Siqi Cai** [1]  **Yuzheng Cai** [4]  **Chaoyou Fu** [2]  **Ke Li** [1]  **Xing Sun** [1]

## Abstract

Computer-using agents (CUAs) enable task completion through natural interaction with operating systems and software interfaces. While script-based verifiers are widely adopted for evaluation, they suffer from limited scalability and inability to provide step-wise assessment. Reward models offer promising alternatives, but their effectiveness on CUA evaluation remains largely underexplored. To address this gap, we present **CUARewardBench**, comprising four key contributions: **(1) First-ever Comprehensive CUA Reward Benchmark:** We introduce the first benchmark for evaluating both outcome reward models (ORM) and process reward models (PRM) on CUA tasks, enabling systematic assessment across trajectory-level and step-level evaluation. **(2) Diverse, Practical and Reliable Dataset:** CUARewardBench encompasses trajectories from 10 software categories and 7 agent architectures with varying performance levels (25.9%-50.8% success rates). All trajectories are expertly annotated through carefully designed protocols, with rigorous quality control to ensure reliability and practical applicability. **(3) Comprehensive Analysis and Insights:** Through extensive experiments across 12 vision-language/reward models and 3 prompt templates, we reveal critical limitations of current CUA RMs, including insufficient visual reasoning capabilities, knowledge deficiencies, and the superiority of general VLMs over specialized CUA models for reward evaluation. **(4) Unanimous Prompt Ensemble (UPE):** Based on the insights from our comprehensive analysis, we propose UPE, a novel ensemble method that significantly enhances reward model reliability through strict unanimous voting and strategic prompt-template configurations. UPE achieves 88.0% precision and 95.3% NPV for ORM, and 83.1% precision and 86.2% NPV for PRM, substantially outperforming single VLMs and traditional ensemble approaches. In short, this work introduces **both a comprehensive benchmark and a novel ensemble method** that substantially enhances CUA reward model reliability. Code is publicly available at `https://github.com/Tencent/CUARewardBench`.

*Equal contribution [1]Tencent Youtu Lab [2]Nanjing University [3]Peking University [4]Fudan University. Correspondence to: Haojia Lin <haojialin@tencent.com>.

*Proceedings of the 43$^{rd}$ International Conference on Machine Learning*, Seoul, South Korea. PMLR 306, 2026. Copyright 2026 by the author(s).

## 1. Introduction

Computer-using agents (CUAs) represent a significant advancement in artificial intelligence, enabling large language models to interact directly with operating systems and software interfaces to accomplish complex tasks autonomously. Recent advances from contemporary agents including OpenAI's Operator (OpenAI, 2025b) and UITARS-2 (Wang et al., 2025a) have demonstrated promising performance across diverse desktop scenarios.

Evaluating CUA performance presents unique challenges that extend beyond traditional language model assessment. While OSWorld benchmark (Xie et al., 2024) initially employed manually predefined scripts for trajectory verification, this approach incurs prohibitively high costs of manual annotation and is proved inadequate for scaling-up. Consequently, VLM-based reward models (RMs) have emerged as a cost-effective alternative for trajectory evaluation. The growing demand for CUA reward models stems from two critical needs: trajectory filtering to identify successful executions for off-line expert imitation as cold-start (Ye et al., 2025; Tang et al., 2025; Sun et al., 2025; Wang et al., 2025b), and providing reward signals for online agent reinforcement learning (RL) (Sun et al., 2025; Wang et al., 2025a). These models must evaluate both outcome success (whether the agent accomplished the task) and stepwise correctness (whether individual actions contribute to the goal). For a comprehensive review of related work, see Section A.1.

However, the effectiveness of existing reward models for computer-using agents remains largely unverified. While several recent works have proposed VLM-based reward models for CUA evaluation (Ye et al., 2025; Tang et al., 2025; Wang et al., 2025a), many lack open-source implementations or detailed methodological descriptions, hindering systematic assessment and community-wide exploration of the capabilities of CUA RMs. This gap underscores the urgent need for standardized benchmarks that can rigorously evaluate and advance computer-using agent reward models.

To address these challenges, we present a systematic investigation into computer-using agent reward models through both benchmark construction and comprehensive evaluation. Our investigation comprises three complementary components. **First, we establish a rigorous evaluation framework by constructing CUARewardBench**, a benchmark specifically tailored to the unique requirements of CUA reward modeling (Figure 2). The benchmark design prioritizes three key aspects: *ecological validity* through diverse desktop software interactions that reflect real-world agent deployment scenarios, *comprehensive coverage* by incorporating trajectories from agents with varying architectural paradigms and capability levels, and *multi-granularity assessment* enabling evaluation of both trajectory-level outcomes and step-level correctness. **Second, we conduct systematic empirical analysis** to identify the critical factors determining reward model effectiveness, characterize their failure modes through detailed error analysis, and explore ensemble strategies that can enhance model reliability by leveraging complementary strengths of different approaches. These analyses reveal fundamental limitations in current approaches and provide insights into the design space for more reliable reward models. **Third, building upon these findings, we propose Unanimous Prompt Ensemble (UPE)**, a novel ensemble method that significantly enhances reward model reliability through strategic unanimous voting and diversified prompt-template configurations (Figure 1). As demonstrated in our experiments, UPE achieves substantial improvements over both single VLMs and traditional ensemble approaches, addressing the critical limitations identified through our empirical analysis. This integrated approach—from benchmark construction to empirical analysis to method development—enables us to provide both a standardized evaluation testbed for the community and an immediately deployable solution for enhancing CUA reward model reliability. The main contributions of this paper are summarized as follows:

- **First-ever Comprehensive CUA Reward Benchmark:** We propose CUARewardBench, the first comprehensive benchmark specifically designed for evaluating both ORM and PRM on CUA trajectories. The benchmark comprises 272 trajectory success annotations and 346 step correct-

ness annotations (Table 2), providing a rigorous testbed for systematic assessment of reward model capabilities across both trajectory-level and step-level verification.

- **Diverse, Practical and Reliable Dataset:** CUAReward-Bench exhibits three key strengths that establish it as a rigorous evaluation framework. First, *diversity*: comprehensive task coverage across 10 software categories and trajectories from 7 distinct policy models with varying capabilities (Section 2.2). Second, *practicality and challenge*: carefully designed protocols for trajectory selection, key step identification, and annotation standards that capture realistic failure modes and critical decision points. Third, *reliability*: extensive human validation and multi-stage quality control to ensure annotation consistency and practical applicability (Section 2.3).

- **Comprehensive Analysis and Insights:** Through extensive experiments across 12 state-of-the-art vision-language/reward models and 3 prompt templates, we reveal that verification asymmetry challenges are significantly weakened in CUA tasks, with visual reasoning capability emerging as the overwhelmingly critical factor that dominates specialized training approaches (Section 3). Error analysis of 53 failure cases identifies reasoning errors (35.8%) and visual understanding errors (30.2%) as primary failure modes, providing actionable insights for future development (Section A.8).

- **Unanimous Prompt Ensemble (UPE):** We propose UPE, a novel ensemble method that significantly enhances reward model reliability for CUA tasks through strict unanimous voting and strategic prompt-template configurations (Section 3.4). As demonstrated in Figure 1, UPE achieves 88.0% precision and 95.3% NPV for ORM, and 83.1% precision and 86.2% NPV for PRM, substantially outperforming single VLMs with different prompts and traditional ensemble methods such as majority voting. This contribution provides a practical and immediately deployable solution for improving the reliability of reward-based CUA training pipelines.

## 2. CUARewardBench

### 2.1. Problem Formulation

**Trajectory Definition.** Let $q$ denote the instruction given by user, $o_i$ denote a system state observation at step $i$, and $a_i$ represent an executable action in an operating environment $\mathcal{E}$ such that $o_{i+1} = \mathcal{E}(o_i, a_i)$. A computer-using agent trajectory is defined as the sequence:

$$\mathcal{T} = \{q, o_1, (r_1, a_1), o_2, (r_2, a_2), \ldots, (r_{n-1}, a_{n-1}), o_n\} \tag{1}$$

where $r_i$ is the agent's reasoning for action $a_i$, and $o_n$ is the terminal state. Each state observation $o_i$ contains a

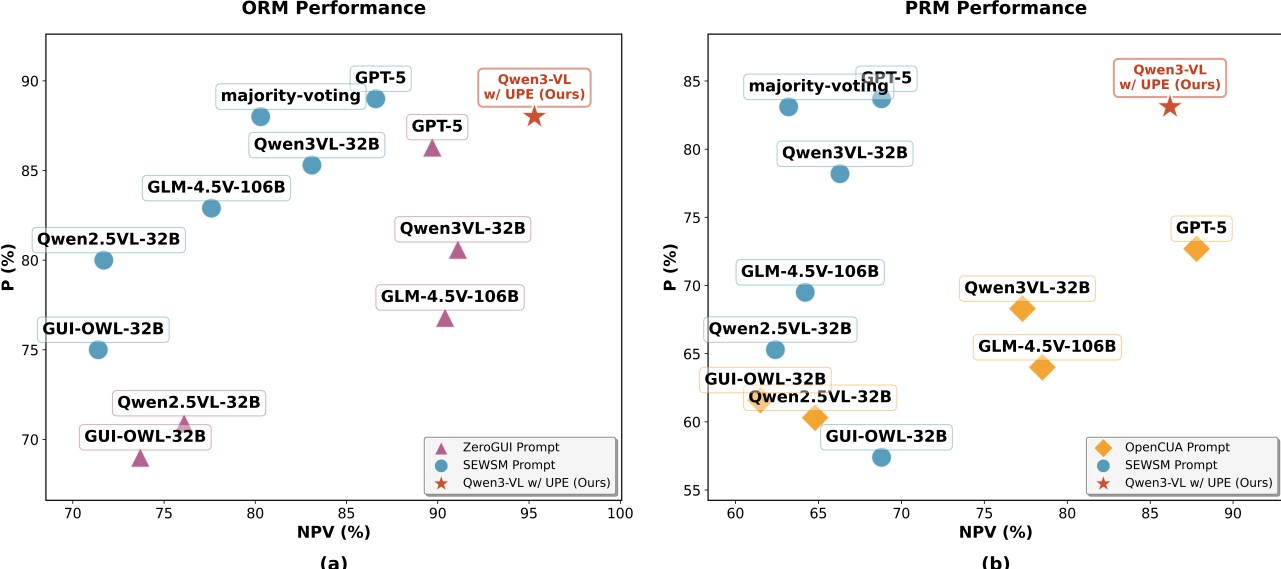

*Figure 1.* **UPE achieves superior reward model reliability.** Performance comparison on ORM and PRM tasks shows that our proposed UPE (red star) simultaneously achieves high precision and NPV, significantly outperforming single VLMs with different prompts and traditional ensemble methods (majority-voting). The upper-right positioning demonstrates UPE's effectiveness in balancing positive and negative prediction accuracy. Details of UPE are discussed in Section 3.4.

**single RGB screenshot** of the current interface, consistent with current reward model practices that use visual inputs exclusively.

**Reward Model Formulation.** Given trajectory $\mathcal{T}$, a reward model $\mathcal{R}$ predicts:

$$\hat{\mathcal{A}} = \mathcal{R}(\mathcal{T}) = (\hat{s}, \{\hat{c}_1, \ldots, \hat{c}_{n-1}\}) \qquad (2)$$

where $\hat{s} \in \{0, 1\}$ indicates trajectory success ($1 =$ success, $0 =$ failure), and $\hat{c}_i \in \{0, 1\}$ denotes step correctness ($1 =$ correctness, $0 =$ non-correctness). In recent researches (Yang et al., 2025a; Sun et al., 2025; Tang et al., 2025; Ye et al., 2025; Wang et al., 2025a), $\mathcal{R}$ is exclusively implemented using VLMs as the core evaluation engine.

The following sections detail the construction of CUARewardBench: Section 2.2 describes the curation process for trajectories $\mathcal{T}$, while Section 2.3 presents the ground truth annotation methodology for $\hat{s}$ and $\hat{c}_i$.

### 2.2. Trajectory Collection.

**Tasks and Environments.** We build CUARewardBench upon OSWorld (Xie et al., 2024), a widely-adopted benchmark that provides comprehensive evaluation environments for computer-using agents across diverse desktop applications. CUARewardBench covers all 10 task categories to provide comprehensive evaluation across the complete spectrum of computer-use scenarios. To ensure benchmark quality and maintain evaluation reliability, we systematically exclude tasks marked as infeasible in the original OSWorld dataset to avoid introducing evaluation noise from inherently

unsolvable scenarios.

**Policy Model Pool.** To ensure trajectory diversity and comprehensive coverage of agent capabilities, we employ 7 distinct CUA models with 10 different configurations (some agents vary in maximum step limits). As shown in Table 1, our agent selection follows two key principles: 1) *Architectural Diversity*: We include both single-model approaches and agentic frameworks to capture different decision-making paradigms. 2) *Performance Spectrum*: Our selected agents span success rates from 25.9% to 50.8% on OSWorld, ensuring comprehensive coverage of capability levels.

This dual consideration ensures trajectory diversity by capturing a wide spectrum of decision-making patterns and failure modes, providing a robust foundation for evaluating reward model generalization. Additionally, our diverse agent pool enhances task coverage for successful trajectories, achieving an oracle success rate of 72.3% and establishing a solid data foundation for subsequent trajectory curation.

**Trajectory Selection Criteria.** Building upon the established agent configurations, we leverage pre-collected trajectories [1] from OSWorld-verified (Xie et al., 2025b) across all 10 task categories and 7 agent models. Our trajectory curation follows systematic criteria designed to ensure benchmark quality and evaluation comprehensiveness:

---

[1] https://huggingface.co/datasets/xlangai/ubuntu_osworld_verified_trajs

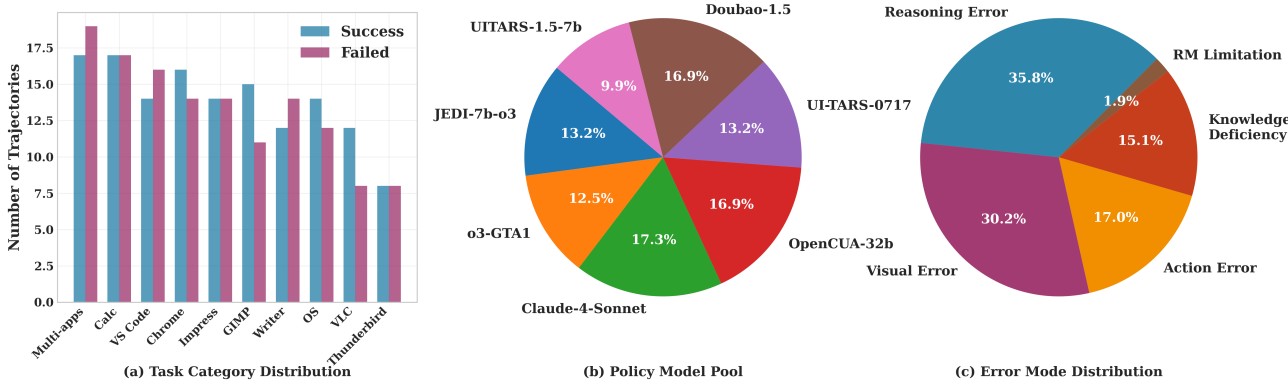

*Figure 2.* CUARewardBench dataset characteristics and selected experimental findings: (a) Task distribution of trajectory annotations across 10 software categories (Section 2.3), (b) policy model diversity in our benchmark (Section 2.2), and (c) key error modes identified in our evaluation experiments (Appendix A.8).

| Agent Model | Arch | SR-15 | SR-50 | Traj |
|---|---|---|---|---|
| JEDI-7b-o3 | Framework | 42.4 | 50.8 | 36 |
| o3-GTA1 | Framework | - | 48.8 | 34 |
| Claude-4-Sonnet | Single | 31.3 | 44.0 | 47 |
| OpenCUA-32b | Single | 28.3 | 33.9 | 46 |
| UI-TARS-0717 | Single | 31.9 | - | 36 |
| Doubao-1-5-Thinking | Single | 28.3 | - | 46 |
| UITARS-1.5-7b | Single | 25.9 | - | 27 |
| **Oracle / Total** | - | | **72.3** | **272** |

*Table 1.* Policy model pool: architecture (Arch), OSWorld success rates (SR) under 15/50-step limits, and trajectory counts (Traj). Refs: JEDI-7b-o3 (Xie et al., 2025a), o3-GTA1 (Yang et al., 2025b), Claude-4-Sonnet (Anthropic, 2025), OpenCUA-32b (Wang et al., 2025b), UI-TARS (Qin et al., 2025), Doubao-1-5-Thinking (Guo et al., 2025).

| Task Category | Succ. | Fail. | Good | Bad |
|---|---|---|---|---|
| Multi-apps | 17 | 19 | 23 | 22 |
| Calc | 17 | 17 | 20 | 17 |
| VS Code | 14 | 16 | 23 | 19 |
| Chrome | 16 | 14 | 24 | 24 |
| Impress | 14 | 14 | 20 | 16 |
| GIMP | 15 | 11 | 20 | 16 |
| Writer | 12 | 14 | 15 | 14 |
| OS | 14 | 12 | 13 | 15 |
| VLC | 12 | 8 | 16 | 10 |
| Thunderbird | 8 | 8 | 8 | 11 |
| **Total** | **139** | **133** | **182** | **164** |

*Table 2.* Annotation distribution across task categories: trajectory counts (Succ./Fail.) and action counts (Good/Bad).

- *Task Category Balance:* We maintain balanced distribution across software categories to enable comparative analysis of reward model performance.

- *Success-Failure Balance:* We ensure proportional coverage of successful and failed trajectories to evaluate RMs' discriminative capabilities across both outcome types.

- *Difficulty Control:* We exclude trajectories where no agent succeeds (too difficult) or where 8+ agent configurations succeed (too easy), ensuring moderate difficulty levels that provide meaningful evaluation challenges.

- *Step Count Constraint:* We select trajectories containing fewer than 25 steps, as this threshold accommodates most task completions while maintaining manageable annotation costs.

### 2.3. Annotation

**Trajectory Success.** Although OSWorld provides script-based trajectory success evaluation, we employ human annotation to ensure annotation reliability and accuracy. Annotators evaluate trajectory success based on two primary criteria:

- *Instruction Consistency*: Whether the agent successfully transitions the computer's final state to match the requirements specified in the instruction.

- *Harmful Side Effects*: Whether the agent causes unintended changes to the computer's final state that are not required by the instruction and would require additional corrective actions to resolve. Note that redundant but harmless operations (e.g., extra clicks on desktop) are not considered violations of this criterion.

**Step Correctness.** We evaluate step correctness based on a single criterion: whether the step's execution positively contributes to trajectory success. Formally, let $o_{\text{ts}}$ denote the trajectory success event. An action $a_i$ is considered correct if $p(o_{\text{ts}}|o_{i+1}, a_i) > p(o_{\text{ts}}|o_i)$. Conversely, for incorrect steps, $p(o_{\text{ts}}|o_{i+1}, a_i) < p(o_{\text{ts}}|o_i)$, indicating that additional actions are required to mitigate the negative effects of $a_i$.

For redundant actions, $p(o_{\text{ts}}|o_{i+1}, a_i) = p(o_{\text{ts}}|o_i)$. However, during our annotation process, we observed that redundant actions cause significantly less harm than incorrect actions, while being considerably more difficult to identify reliably. To enhance annotation objectivity, we focus primarily on distinguishing between correct and incorrect actions, largely excluding redundant actions from our evaluation. Furthermore, we do not annotate every action in the trajectories selected from Section 2.2. Instead, we focus on steps that are critical to trajectory success. Formally, we define two types of key actions:

- *Key Good Actions*: Actions where $p(o_{\text{ts}}|o_{i+1}, a_i) - p(o_{\text{ts}}|o_i)$ is as large as possible and $p(a_i|o_i)$ is as small as possible. Intuitively, these actions significantly advance task completion while being hard to infer.

- *Key Bad Actions*: Actions where $|p(o_{\text{ts}}|o_{i+1}, a_i) - p(o_{\text{ts}}|o_i)|$ is as large as possible and $p(a_i|o_i)$ is as large as possible. These represent actions that substantially hinder task success while appearing deceptively reasonable.

**Annotation Statistics.** As shown in Table 2, our final dataset comprises 139 successful and 133 failed trajectories for trajectory-level evaluation, and 182 good and 164 bad actions for action-level assessment across all task categories. For trajectory-level annotations, although OSWorld (Xie et al., 2024) provides script-based trajectory success evaluation, we employ human annotation to ensure annotation reliability and maintain high-quality standards. For action-level annotations, we selectively annotate key actions following the criteria outlined in the previous section, focusing on steps that are most critical for trajectory success evaluation. Notably, we further annotate bad actions within successful trajectories and good actions within failed trajectories, recognizing that trajectory success and step correctness are orthogonal. This enables robust evaluation of reward models across varying trajectory contexts.

## 3. Reward Performance and Analysis

Reward model implementation involves two key dimensions: VLM and prompt. Our CUARewardBench systematically evaluates both to provide comprehensive insights into CUA RM.

### 3.1. VLM Selection As Reward Models

**VLM Selection.** We primarily evaluate open-source models to facilitate reproducibility and large-scale deployment in practical applications such as data construction and online reinforcement learning training. We assess four model categories: General VLMs, Visual Reasoning Models, Specialized CUA Models, and Specialized CUA Reward Models.

| Metric | Formula | Interpretation | Aspect |
|---|---|---|---|
| Precision | $\frac{TP}{TP+FP}$ | Pred. positives that are actually positive | reliability |
| NPV | $\frac{TN}{TN+FN}$ | Pred. negatives that are actually negative | reliability |
| Recall | $\frac{TP}{TP+FN}$ | Actual positives that are correctly predicted | efficiency |
| Specificity | $\frac{TN}{TN+FP}$ | Actual negatives that are correctly predicted | efficiency |

*Table 3.* Evaluation metrics for CUA reward models. TP/FP/TN/FN = True/False Positive/Negative. Aspect: reliability = reward reliability, efficiency = sample efficiency.

Detailed model selection and specifications are provided in Appendix A.3.

**Prompt Template.** Among existing works, we identify three that have open-sourced their prompt templates for CUA reward modeling:

- *ORM prompt of ZeroGUI* (Yang et al., 2025a): ZeroGUI designs a frame-by-frame captioning followed by holistic analysis prompt template, requiring Qwen2.5VL-32B to evaluate whether trajectories accomplish their tasks. We adopt ZeroGUI's prompt template as CUARewardBench's ORM prompt template.

- *Step reflector prompt of OpenCUA* (Wang et al., 2025b): OpenCUA employs Claude as a reflector in their chain-of-thought data annotation pipeline, generating reflections for current steps based on previous step reasoning and current screenshots. Considering the coupling complexity between the reflector and previous step reasoning, and to ensure fair comparative experiments, we simplify the reflector prompt as CUARewardBench's PRM prompt template. The detailed comparison before and after simplification is provided in Section A.4.

- *Prompt of SE-WSM* (Sun et al., 2025): SE-WSM conducts step-by-step analysis of input trajectories, providing multi-dimensional evaluations including trajectory correctness, redundant steps, first error step, and correct action suggestions. This comprehensive evaluation covering both coarse and fine-grained assessments enables it to function as both ORM and PRM. We adopt SE-WSM's prompt template for both ORM and PRM evaluation in CUARewardBench.

Detailed prompt templates and model output parsing methods are provided in Section A.4 .

**Evaluation Metrics.** Existing frameworks use different metrics: AgentRewardBench (Lu et al., 2025) focuses on precision, while SE-WSM (Sun et al., 2025) uses precision

and NPV. We consider two use cases: offline trajectory filtering and online RL reward provision. For offline filtering, *precision* measures reward reliability (correctly identified successful trajectories), while *recall* measures sample efficiency (coverage of actual positives). For online RL, where negative samples also contribute to model updates, we introduce *NPV* (reliability of negative signals) and *specificity* (sample efficiency for negatives) as counterparts to precision and recall. Table 3 summarizes our four complementary metrics. Since reward reliability is more critical than sample efficiency in both scenarios, we prioritize precision and NPV as primary metrics, with recall and specificity as secondary indicators.

### 3.2. Effect of VLM Selection

**Model size and training quality comprehensively impact reward model performance.** 1) Across all evaluated models—including general VLMs, specialized CUA models VLMs, and specialized CUA reward models—7B variants consistently underperform their 32B+ counterparts across different metrics. While CUA-specific training enhances the reward prediction capabilities of 7B models (e.g., GUI-OWL-7B vs. Qwen2.5VL-7B in Table 4 and Table 5), they still lag behind larger models. 2) Surprisingly, Qwen2.5VL-72B underperforms compared to the 32B variant, with particularly notable differences in action-level evaluation. A possible explanation is that Qwen2.5VL-32B received additional RL training compared to the 72B model[2], which improved its reasoning capabilities and consequently led to stronger generalization for CUA reward prediction.

**Visual reasoning capability is the core element of CUA reward models.** GUI-OWL-32B, despite being post-trained from Qwen2.5VL-32B specifically for CUA tasks, consistently underperforms its base model across all prompt configurations. Through detailed response analysis, we observe that GUI-OWL-32B produces significantly shorter reasoning processes compared to Qwen2.5VL-32B, indicating that despite incorporating general reasoning data during post-training, the model still experiences some degradation in reasoning capabilities. The strong performance of visual reasoning models such as GLM-4.5V-106B, Qwen3VL, GPT-5, Qwen3.5, and Kimi-K2.5 further corroborates this finding, indicating that visual reasoning remains a central capability for CUA reward evaluation.

**CUA policy training benefits reward evaluation capabilities, but only when reasoning abilities are preserved.** GUI-OWL exhibits a performance paradox: GUI-OWL-7B significantly outperforms Qwen2.5VL-7B, while GUI-OWL-32B shows notable degradation compared to Qwen2.5VL-32B. This counterintuitive pattern stems from the differential impact of CUA specialization on mod-

els with varying baseline reasoning capabilities. For Qwen2.5VL-7B, which possesses inherently weak reasoning abilities, the benefits gained from CUA training outweigh the negative effects of reasoning capability degradation. Conversely, for Qwen2.5VL-32B with strong baseline reasoning capabilities, the negative impact of reasoning degradation overshadows the gains from CUA training. This finding suggests that CUA policy training can enhance reward evaluation performance, but only when the specialization process preserves the model's fundamental reasoning abilities.

**Specialized CUA reward models require diverse training data to achieve effective generalization.** SE-WSM (Sun et al., 2025), a specialized CUA reward model fine-tuned from Qwen2.5VL-7B, shows overall performance on CUARewardBench that is comparable to, but slightly worse than, its Qwen2.5VL-7B base model. This underperformance likely stems from its narrowly scoped training data coverage, which comprises only 860 trajectories from 43 Chrome tasks. While this narrow scope may suffice for web-only benchmarks like AgentRewardBench (Lu et al., 2025), it proves inadequate for generalizing to the comprehensive task categories covered in our CUARewardBench. The comprehensive task-category coverage and policy model diversity in CUARewardBench successfully expose limitations that remain hidden in more limited evaluation settings, demonstrating its effectiveness as a rigorous testbed for CUA RM capabilities.

### 3.3. Impact of Prompt Templates

**Prompt templates primarily influence P-NPV trade-offs rather than overall performance improvements.** As shown in Tables 4 and 5, different prompt templates create distinct evaluation standards that affect the precision-recall balance. Taking Qwen2.5VL-32B as an example, in ORM settings, the *sewsm* prompt achieves 9.1 percentage points higher precision than *zerogui*, but suffers a 4.4 percentage point drop in NPV. This trade-off pattern is consistently observed in PRM settings, where *sewsm* outperforms *open-cua_reflector* by 5.0 percentage points in precision while losing 2.4 percentage points in NPV.

**SE-WSM prompt enforces stricter trajectory success criteria compared to ZeroGUI.** Through comparative analysis of prompts and model responses, we find that *sewsm* employs stricter success standards than *zerogui*. The *sewsm* template requires VLMs to verify trajectory reasonableness across multiple dimensions, including trajectory correctness, redundant steps, first error identification, and correct action suggestions. These additional evaluation dimensions naturally increase the likelihood that successful trajectories may be incorrectly rejected due to minor imperfections. Conversely, *zerogui* adopts relaxed criteria, requiring only

---

[2]https://huggingface.co/Qwen/Qwen2.5-VL-32B-Instruct

| Reward Model | Overall | | | | | vscode | | gimp | | writer | | chrome | | multi_apps | |
|---|---|---|---|---|---|---|---|---|---|---|---|---|---|---|---|
| | P | NPV | R | S | OA | P | NPV | P | NPV | P | NPV | P | NPV | P | NPV |
| *zerogui* | | | | | | | | | | | | | | | |
| Qwen2.5VL-7B | 60.8 | 65.0 | 74.8 | 49.2 | 62.1 | 47.8 | 57.1 | 57.1 | 40.0 | 46.7 | 54.5 | 72.2 | 72.7 | 93.3 | 85.7 |
| Qwen2.5VL-32B | 70.9 | 76.1 | 80.6 | 65.2 | 72.8 | 75.0 | 85.7 | 68.8 | 60.0 | 53.3 | 63.6 | 70.0 | 80.0 | 85.0 | **100.0** |
| Qwen2.5VL-72B | 70.0 | 72.1 | 75.5 | 66.2 | 71.0 | 65.0 | 90.0 | 70.6 | 66.7 | 60.0 | 62.5 | 66.7 | 77.8 | 88.9 | 94.4 |
| GLM-4.5V-106B | 76.8 | 90.4 | **92.8** | 70.7 | 82.0 | 73.7 | **100.0** | 75.0 | **100.0** | 66.7 | 81.8 | 72.7 | **100.0** | 89.5 | **100.0** |
| Qwen3VL-32B-T | 80.6 | **91.1** | **92.8** | 76.7 | 84.9 | **85.7** | 87.5 | 72.2 | 75.0 | **78.6** | **91.7** | 80.0 | **100.0** | 94.1 | 94.7 |
| Qwen3VL-235B-T | 82.6 | 90.6 | 92.1 | 79.7 | 86.0 | 82.4 | **100.0** | 76.5 | 77.8 | 66.7 | 81.8 | **93.8** | 92.9 | 94.1 | 94.7 |
| GPT-5 | **86.3** | 89.7 | 90.6 | **85.0** | **87.9** | 81.2 | 92.9 | **81.2** | 80.0 | **78.6** | **91.7** | 88.2 | 92.3 | 94.1 | 94.7 |
| GUI-OWL-7B | 69.0 | 68.8 | 71.9 | 65.6 | 68.4 | 70.6 | 83.3 | 61.9 | 60.0 | 64.3 | 75.0 | 66.7 | 77.8 | 88.2 | 88.9 |
| GUI-OWL-32B | 69.0 | 73.7 | 78.4 | 63.2 | 71.0 | 64.3 | 68.8 | 65.0 | 66.7 | 61.5 | 69.2 | 73.7 | 81.8 | **100.0** | **100.0** |
| *sewsm* | | | | | | | | | | | | | | | |
| Qwen2.5VL-7B | 63.1 | 57.1 | 50.4 | 69.2 | 59.6 | 50.0 | 54.5 | 66.7 | 47.1 | 46.7 | 54.5 | 70.6 | 69.2 | 84.6 | 73.9 |
| Qwen2.5VL-32B | 80.0 | 71.7 | 69.1 | 82.0 | 75.4 | 84.6 | 82.4 | 75.0 | 57.1 | 63.6 | 66.7 | 75.0 | 71.4 | 87.5 | 85.0 |
| Qwen2.5VL-72B | 78.6 | 74.5 | 74.1 | 78.9 | 76.5 | 85.7 | 87.5 | 78.6 | 66.7 | 54.5 | 60.0 | 78.9 | 90.9 | **100.0** | 86.4 |
| GLM-4.5V-106B | 82.9 | 77.6 | 77.0 | 83.5 | 80.1 | 78.6 | 81.2 | 75.0 | 70.0 | 69.2 | 76.9 | 84.2 | **100.0** | 92.9 | 81.8 |
| Qwen3VL-32B-T | 85.3 | 83.1 | 83.5 | 85.0 | 84.2 | 80.0 | 86.7 | 78.6 | 66.7 | 78.6 | **91.7** | 87.5 | 85.7 | 93.3 | 85.7 |
| Qwen3VL-235B-T | 85.7 | 78.6 | 77.7 | 86.4 | 81.6 | 80.0 | 86.7 | 75.0 | 57.1 | 80.0 | 75.0 | **93.8** | 92.9 | 87.5 | 85.0 |
| GPT-5 | 89.0 | 86.6 | 87.1 | 88.5 | 87.1 | 92.9 | 93.8 | 82.4 | **88.9** | **92.3** | **100.0** | 86.7 | 80.0 | 94.1 | 94.7 |
| Qwen3.5-397B-A17B-FP8 | **89.3** | 84.4 | 84.2 | 89.5 | 86.8 | 86.7 | 93.3 | **92.3** | 76.9 | 66.7 | 71.4 | **94.1** | **100.0** | 94.1 | 94.7 |
| Kimi-K2.5 | 88.4 | **92.1** | **92.8** | 87.2 | **90.1** | 82.4 | **100.0** | 81.2 | 80.0 | 91.7 | 92.9 | 88.9 | **100.0** | 94.1 | 94.7 |
| GUI-OWL-7B | 68.4 | 72.4 | 76.8 | 63.2 | 69.9 | 66.7 | 66.7 | 64.7 | 55.6 | 52.6 | 71.4 | 76.2 | **100.0** | 84.2 | 94.1 |
| GUI-OWL-32B | 75.0 | 71.4 | 71.2 | 75.2 | 73.2 | 70.0 | 65.0 | 68.8 | 60.0 | 90.0 | 81.2 | 77.8 | 83.3 | 85.0 | **100.0** |
| SE-WSM-7B | 70.0 | 52.2 | 20.1 | **91.0** | 54.8 | **100.0** | 57.1 | 62.5 | 44.4 | 66.7 | 56.5 | 88.9 | 61.9 | **100.0** | 55.9 |
| *voting-majority* | | | | | | | | | | | | | | | |
| G106-s 2runs | 84.3 | 70.7 | 65.5 | 87.2 | 76.1 | 76.9 | 76.5 | 71.4 | 58.3 | 70.0 | 68.8 | 88.2 | 92.3 | **100.0** | 76.0 |
| Q32-s + G106-s | **90.1** | 68.5 | 59.0 | **93.2** | 75.7 | **90.9** | 78.9 | **87.5** | 55.6 | 62.5 | 61.1 | 92.3 | 76.5 | 92.3 | 78.3 |
| Q32-s + G106-s + G106-z | 81.6 | **84.8** | **86.3** | 79.7 | 83.1 | 80.0 | 86.7 | 73.7 | **85.7** | 68.8 | **90.0** | 72.7 | **100.0** | 93.8 | **90.0** |
| Q3-32-s + Q3-235-z | 88.0 | 80.3 | 79.1 | 88.7 | **83.8** | 85.7 | **87.5** | 76.9 | 61.5 | **76.9** | 84.6 | **92.9** | 81.2 | 92.9 | 81.8 |
| *voting-strict_unanimous* | | | | | | | | | | | | | | | |
| G106-s 2runs | 84.3 | 82.3 | 65.5 | **76.7** | 71.0 | 76.9 | 86.7 | 71.4 | 75.0 | 70.0 | 76.9 | 88.2 | **100.0** | **100.0** | 90.0 |
| Q32-s + G106-s | **90.1** | 84.2 | 59.0 | 72.2 | 65.4 | **90.9** | 85.7 | **87.5** | 83.3 | 62.5 | **90.0** | 92.3 | **100.0** | 92.3 | 89.5 |
| Q32-s + G106-s + G106-z (**w/ UPE**) | 89.8 | 93.3 | 56.8 | 63.2 | 59.9 | **90.9** | **100.0** | **87.5** | **100.0** | 57.1 | **90.0** | 92.3 | **100.0** | 92.3 | **100.0** |
| Q3-32-s + Q3-235-z (**w/ UPE**) | 88.0 | **95.3** | **79.1** | 75.9 | **77.6** | 85.7 | **100.0** | 76.9 | 87.5 | **76.9** | **90.0** | **92.9** | **100.0** | 92.9 | **100.0** |

*Table 4.* Performance comparison of outcome reward models (ORM) across different vision-language models, prompt configurations, and task categories. Results show precision (P), negative predictive value (NPV), recall (R), and specificity (S) for trajectory success evaluation under two prompt configurations (*zerogui* and *sewsm*) and two voting strategies (*voting-majority* and *voting-strict_unanimous*). In model names, "-z" denotes *zerogui* prompt and "-s" denotes *sewsm* prompt; "Q32" denotes Qwen2.5VL-32B, "G106" denotes GLM-4.5V-106B, "Q3-32" denotes Qwen3VL-32B, and "Q3-235" denotes Qwen3VL-235B. Due to space constraints, complete results across all task categories are provided in Table 7.

binary trajectory success determination, which allows some failed trajectories to deceive VLMs through superficial reasonableness.

**OpenCUA reflector adopts more relaxed action evaluation criteria compared to SE-WSM.** The key distinction between *opencua_reflector* and *sewsm* prompts lies in their temporal scope: *opencua_reflector* receives only truncated trajectories up to the current step, limiting its perspective to immediate action effects on current states without global visibility into subsequent steps or overall task impact. This constraint makes the former susceptible to deceptive bad actions that appear reasonable in isolation but prove detrimental to task completion. In contrast, *sewsm* processes complete trajectories, naturally providing stricter constraints for action correctness evaluation through comprehensive temporal context.

### 3.4. Ensemble Methods

Building upon the insights from previous analyses, we propose **Unanimous Prompt Ensemble (UPE)**, a novel ensemble approach that significantly enhances reward model reliability for CUA tasks. UPE integrates two complementary strategies: (1) a strict unanimous voting mechanism that prioritizes prediction reliability over sample efficiency, and (2) strategic prompt-template configurations that leverage the complementary P-NPV trade-offs identified in Section 3.3. Together, these components enable substantial improvements in both precision and negative predictive value, which are critical metrics for ensuring reliable reward signals in reinforcement learning applications.

**Strict Unanimous Voting.** While ZeroGUI (Yang et al., 2025a) employs majority voting, our experiments reveal a critical limitation: it improves precision but substantially

| Reward Model | Overall | | | | | vscode | | gimp | | writer | | chrome | | multi_apps | |
|---|---|---|---|---|---|---|---|---|---|---|---|---|---|---|---|
| | P | NPV | R | S | OA | P | NPV | P | NPV | P | NPV | P | NPV | P | NPV |
| *opencua_reflector* | | | | | | | | | | | | | | | |
| Qwen2.5VL-7B | 54.4 | 49.4 | 53.8 | 50.0 | 52.0 | 53.6 | 42.9 | 59.1 | 50.0 | 54.5 | 50.0 | 44.0 | 43.5 | 63.2 | 57.7 |
| Qwen2.5VL-32B | 60.3 | 64.8 | 79.8 | 41.5 | 61.7 | 58.3 | 66.7 | 76.2 | 73.3 | 60.0 | 66.7 | 57.1 | 69.2 | 62.5 | 76.9 |
| Qwen2.5VL-72B | 58.5 | 64.8 | 83.0 | 34.8 | 60.1 | 64.5 | 72.7 | 73.1 | 90.0 | 54.2 | 60.0 | 52.4 | 66.7 | 56.2 | 61.5 |
| GLM-4.5V-106B | 64.0 | 78.5 | 89.0 | 44.5 | 67.9 | 66.7 | 88.9 | 74.1 | **100.0** | 57.9 | 60.0 | 57.1 | 69.2 | 71.0 | **92.9** |
| Qwen3VL-32B-T | 68.3 | 77.3 | 85.2 | 56.1 | 71.4 | **82.1** | 81.8 | 72.0 | 81.8 | 66.7 | 64.3 | 64.5 | 76.5 | 72.4 | 87.5 |
| Qwen3VL-235B-T | 66.5 | 75.0 | 84.1 | 53.0 | 69.4 | 75.9 | 92.3 | 78.3 | 84.6 | 60.0 | 57.1 | 59.4 | 68.8 | **75.0** | 76.2 |
| GPT-5 | **72.7** | **87.8** | **92.3** | **61.6** | **77.7** | 80.8 | 87.5 | **86.4** | 92.9 | 70.0 | 88.9 | 73.3 | 88.9 | 70.4 | **92.9** |
| GUI-OWL-7B | 64.6 | 61.8 | 67.0 | 59.1 | 63.3 | 63.6 | 55.0 | 66.7 | 60.0 | 62.5 | 61.5 | 55.2 | 57.9 | 66.7 | 62.5 |
| GUI-OWL-32B | 61.6 | 61.5 | 71.4 | 50.6 | 61.6 | 62.5 | 70.0 | 66.7 | 66.7 | 56.2 | 53.8 | 59.4 | 68.8 | 70.0 | 64.0 |
| *sewsm* | | | | | | | | | | | | | | | |
| Qwen2.5VL-7B | 56.7 | 54.2 | 67.0 | 43.3 | 55.8 | 62.5 | 55.6 | 50.0 | 40.9 | 56.2 | 53.8 | 57.1 | 69.2 | 55.6 | 55.6 |
| Qwen2.5VL-32B | 65.3 | 62.4 | 67.8 | 59.8 | 64.0 | 74.1 | 80.0 | 76.5 | 63.2 | 69.2 | 62.5 | 71.4 | 66.7 | 66.7 | 66.7 |
| Qwen2.5VL-72B | 58.7 | 67.1 | 85.2 | 33.5 | 60.7 | 60.0 | 71.4 | 72.7 | 71.4 | 58.3 | 80.0 | 56.4 | 77.8 | 56.2 | 61.5 |
| GLM-4.5V-106B | 69.5 | 64.2 | 66.1 | 67.7 | 66.9 | 68.8 | **90.0** | 81.2 | 65.0 | 72.7 | 61.1 | 80.0 | 71.4 | 63.6 | 60.9 |
| Qwen3VL-32B-T | 78.2 | 66.3 | 63.2 | 80.5 | 71.4 | 85.7 | 76.2 | 91.7 | 62.5 | 78.6 | 73.3 | 70.6 | 61.3 | **90.9** | **80.6** |
| Qwen3VL-235B-T | 74.5 | 63.2 | 59.3 | 77.4 | 67.9 | 77.8 | 86.7 | 92.3 | 65.2 | 57.1 | 53.3 | 90.9 | 62.2 | 66.7 | 59.3 |
| GPT-5 | **83.7** | 68.8 | 64.8 | **86.0** | **74.9** | 90.0 | 77.3 | 92.3 | 65.2 | **88.9** | 65.0 | 94.1 | 74.2 | 80.0 | 63.3 |
| Qwen3.5-397B -A17B-FP8 | 77.0 | 63.8 | 58.8 | 80.5 | 69.1 | 89.5 | 73.9 | 92.3 | 65.2 | 80.0 | 63.2 | 83.3 | 61.1 | 70.6 | 60.7 |
| Kimi-K2.5 | 72.4 | 65.0 | 64.8 | 72.6 | 68.5 | 74.1 | 80.0 | **100.0** | **76.2** | **88.9** | 65.0 | 80.0 | 63.6 | 56.5 | 54.5 |
| GUI-OWL-7B | 56.6 | 64.2 | 86.8 | 26.2 | 58.1 | 59.4 | 60.0 | 59.3 | 55.6 | 54.2 | 60.0 | 53.5 | **80.0** | 52.6 | 57.1 |
| GUI-OWL-32B | 57.4 | **68.8** | **89.0** | 26.8 | 59.5 | 57.9 | 75.0 | 64.3 | 75.0 | 70.0 | **88.9** | 53.8 | 66.7 | 55.0 | 80.0 |
| SE-WSM-7B | 58.7 | 52.0 | 48.4 | 62.2 | 54.9 | 76.2 | 66.7 | 58.3 | 45.8 | 70.0 | 57.9 | 60.0 | 57.1 | 57.9 | 53.8 |
| *voting-majority* | | | | | | | | | | | | | | | |
| G106-s 2runs | 73.6 | 62.8 | 59.6 | 76.2 | 67.4 | 72.4 | **84.6** | 81.2 | 65.0 | 72.7 | 61.1 | 85.7 | 64.7 | 68.8 | 58.6 |
| Q32-s + G106-s | 75.6 | 60.5 | 52.5 | 81.1 | 66.0 | 79.2 | 77.8 | **100.0** | 64.0 | **85.7** | 59.1 | 83.3 | 61.1 | 76.5 | 64.3 |
| Q32-s + G106-s + G106-o | 68.4 | **71.5** | **78.6** | 59.8 | 69.7 | 71.9 | **100.0** | 83.3 | 72.2 | 68.8 | 69.2 | 80.0 | **82.6** | 70.8 | 71.4 |
| Q3-32-s + Q3-235-o | **83.1** | 63.2 | 53.8 | **87.8** | 69.9 | **94.4** | 75.0 | 90.9 | **60.0** | 75.0 | **57.1** | 90.0 | 60.5 | **80.0** | **63.3** |
| *voting-strict_unanimous* | | | | | | | | | | | | | | | |
| G106-s 2runs | 73.6 | 64.2 | **59.6** | **57.9** | **58.8** | 72.4 | 88.9 | 81.2 | 68.4 | 72.7 | 56.2 | 85.7 | 75.0 | 68.8 | 59.1 |
| Q32-s + G106-s | 75.6 | 69.1 | 52.5 | 46.3 | 49.6 | 79.2 | 100.0 | 100.0 | 64.3 | 85.7 | 66.7 | 83.3 | 84.2 | 76.5 | 62.5 |
| Q32-s + G106-s + G106-o (**w/ UPE**) | 81.7 | 85.1 | 48.9 | 24.4 | 37.3 | 81.8 | 100.0 | 100.0 | 100.0 | 83.3 | 62.5 | 87.5 | 100.0 | 86.7 | 85.7 |
| Q3-32-s + Q3-235-o (**w/ UPE**) | **83.1** | **86.2** | 53.8 | 45.7 | 50.0 | **94.4** | 100.0 | 90.9 | 91.7 | 75.0 | **87.5** | 90.0 | 77.8 | 80.0 | **86.7** |

*Table 5.* Evaluation results of process reward models (PRM) for step-level correctness assessment across various vision-language models and prompt configurations. The table presents precision (P), negative predictive value (NPV), recall (R), and specificity (S) metrics under two prompt configurations (*opencua_reflector* and *sewsm*) and two voting strategies (*voting-majority* and *voting-strict_unanimous*). In model names, "-o" denotes *opencua_reflector* prompt and "-s" denotes *sewsm* prompt; "Q32" denotes Qwen2.5VL-32B, "G106" denotes GLM-4.5V-106B, "Q3-32" denotes Qwen3VL-32B, and "Q3-235" denotes Qwen3VL-235B. Due to space constraints, complete results across all task categories are provided in Table 8.

reduces NPV (e.g., Q32-s + G106-s vs. G106-s in Tables 4 and 5). In CUA training, reward reliability is more critical than sample efficiency. Reduced sample efficiency can be compensated by increased sampling, but unreliable rewards directly compromise RL training quality. Therefore, we introduce **strict unanimous voting**. Extending traditional unanimous voting (Kuncheva, 2014), our strategy requires consensus on both positive and negative predictions: a sample is classified only when all ensemble members unanimously agree; otherwise, it is abstained. Table 6 illustrates the distinction with concrete voting scenarios, comparing how strict unanimous voting and majority voting produce different decisions under identical ensemble configurations. As shown in Table 4 and 5, this approach substantially improves both precision and NPV for ORM and PRM. While recall and specificity decrease, they remain acceptable—a favorable trade-off when reliability outweighs coverage.

| Voting Scenario | Majority Voting | Strict Unanimous |
|---|---|---|
| 2 Pos., 0 Neg. | Positive ✓ | Positive ✓ |
| 1 Pos., 1 Neg. | Negative ✕ | No Prediction △ |
| 0 Pos., 2 Neg. | Negative ✕ | Negative ✕ |

*Table 6.* Comparison between majority voting and strict unanimous voting strategies. ✓ indicates positive prediction, ✕ indicates negative prediction, and △ indicates no prediction when consensus cannot be reached.

**Prompt-Template Ensemble.** As revealed in Section 3.3, different prompt templates exhibit distinct P-NPV trade-off characteristics. We exploit this complementarity by strategically combining models configured with diverse prompt templates within our voting ensemble. As shown in Table 4, Q32-s + G106-s +G106-z outperforms Q32-s + G106-s:

while precision decreases slightly by 0.3 percentage points, NPV increases by 9.1 percentage points. This demonstrates that for ORM, this heterogeneous prompt configuration yields further NPV improvements beyond those achieved by strict unanimous voting strategy alone. For PRM, the benefits are even more pronounced (Q32-s + G106-s +G106-o vs. Q32-s + G106-s in Table 5), with substantial simultaneous gains in both precision and NPV. This synergy between strict unanimous voting and prompt diversity establishes UPE as an effective method for enhancing reward model reliability in CUA evaluation.

**Downstream validation.** Beyond benchmark metrics, we further validate that CUARewardBench-guided reward modeling can improve downstream agent training. In an end-to-end SFT experiment, we use a strong benchmark-selected reward model to filter 4,370 LibreOffice Impress rollouts into 1,532 high-quality trajectories. Fine-tuning Qwen3VL-7B-Thinking on these filtered trajectories improves its OSWorld LibreOffice Impress success rate from 34.68% to 46.10%, yielding a 33% relative improvement. This result demonstrates that reward models performing well on CUAReward-Bench can identify training data that transfers into practical agent gains; full details are provided in Appendix A.6.

## Impact Statement

This paper presents work whose goal is to advance the field of Machine Learning, specifically in the area of reward model evaluation for computer-use agents. There are many potential societal consequences of our work, none which we feel must be specifically highlighted here.

Our research focuses on benchmark construction and evaluation methodology, which does not directly involve human subjects or crowdsourcing. The dataset and evaluation framework we propose aim to improve the reliability of AI agent systems, which could contribute to safer and more trustworthy AI deployments.

## Acknowledgments

The contribution of Chaoyou Fu was supported by the National Natural Science Foundation of China (Grant No. 62506158 and No. 62441234), the Basic Research Program of Jiangsu (BK20251183), and the CCF-Tencent Rhino-Bird Open Research Fund.

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

# A. Appendix

## A.1. Related Work

**Computer-Use Agents.** Computer-use agent approaches can be broadly categorized into three methodological paradigms. Text-based language models leverage structured GUI metadata such as DOM trees and accessibility labels to generate symbolic commands, ranging from early page-centric agents (Nakano et al., 2021) to recent language-only planners that avoid raw pixel processing (Xu et al., 2023). Vision-centric agents incorporate screen imagery through two main strategies: grounding-focused methods that learn to associate natural-language references with bounding boxes or coordinate clicks (Gou et al., 2024; Wu et al., 2024; Xie et al., 2025a), and end-to-end policies that directly translate screenshots into action sequences (Xu et al., 2024; Qin et al., 2025; Guo et al., 2025). Agent frameworks represent a third paradigm that enhances large language models with specialized components including vision encoders, hierarchical or search-based planners, episodic memory, and tool APIs to tackle long-horizon tasks requiring integrated perception, reasoning, and control (Agashe et al., 2024; 2025). Complementary to these framework-level designs, retrieval-augmented mobile agents incorporate external knowledge acquisition to improve decision making in interactive environments (Li et al., 2026).

**Reward Models for CUA.** OSWorld benchmark (Xie et al., 2024) initially employed manually predefined scripts to verify agent trajectory success. However, writing custom verification scripts for each task incurs prohibitively high costs, making this approach inadequate for large-scale training datasets or online reinforcement learning systems. Consequently, recent approaches leverage Vision-Language Models (VLMs) as reward models for trajectory verification. These reward model applications can be categorized into two primary paradigms. The first paradigm focuses on *trajectory filtering*, where reward models identify successful trajectories. SEAgent (Sun et al., 2025) trains a world state model to determine trajectory success and identify the first error step. OpenCUA (Wang et al., 2025b) employs Claude-3.7 for step-by-step reflection, ultimately judging whether trajectories accomplish their assigned tasks. GUI-OWL (Ye et al., 2025) utilizes both LLMs and VLMs to generate two-channel step-level critics, aggregating these assessments to determine trajectory success. SEA (Tang et al., 2025) develops a step filtering model to remove erroneous and redundant trajectories from training data. The second paradigm employs reward models to *provide reward signals for reinforcement learning*. UITARS-2 (Wang et al., 2025a) uses the UITARS-2 model itself as an outcome reward model (ORM) for general web task verification in RL settings. SEAgent (Sun et al., 2025) applies adversarial imitation punishment to first error steps identified by their world state model. Beyond CUA-specific settings, recent work has also explored automated training of universal process reward models through ensemble prompting and reverse verification (Tan et al., 2026), highlighting the broader relevance of scalable process-level verification.

**Reward Benchmarks.** There is some research evaluating reward models across multiple domains. RewardBench (Lambert et al., 2024) established multi-domain evaluation for LLMs covering chat, reasoning, and safety. RM-Bench (Liu et al., 2024) introduced Best-of-N evaluation, while Multimodal RewardBench (Yasunaga et al., 2025) proposed evaluation frameworks for VLMs with expert-annotated triplets. For agents, AgentRewardBench (Lu et al., 2025) evaluates Web agents but ignores desktop operations, while Agent-RewardBench (Men et al., 2025) covers multimodal agents but lacks CUA-specific capabilities like GUI positioning accuracy. In contrast, our CUARewardBench investigates reward model capabilities specifically for computer-using agents, covering desktop software operations and multi-step decision-making that previous benchmarks do not possess.

## A.2. Annotator Workflow

To ensure annotation quality and reliability, we implement a rigorous dual-phase annotation process with comprehensive quality control measures:

**Phase 1: Independent Annotation.** Each trajectory is systematically annotated by expert annotators following the protocols outlined above. Annotators provide detailed justifications for all annotations, particularly for failed trajectories and incorrect actions. These justifications document the specific reasons for failure, such as incorrect element selection, improper action execution, or deviation from task requirements. This documentation serves both as quality assurance and as valuable context for subsequent cross-validation.

**Phase 2: Cross-Validation and Consensus.** All annotations undergo thorough cross-validation by independent reviewers. For cases with annotation disagreements, the two annotation teams engage in comprehensive discussions, jointly reviewing the original justifications, re-examining the trajectory execution, and analyzing the task requirements in detail. Through this collaborative deliberation process, most disagreements are successfully resolved with consensus. However, cases where

consensus cannot be definitively reached after thorough discussion—typically involving ambiguous task specifications or borderline execution quality—are excluded from the final benchmark to maintain annotation reliability. This conservative approach prioritizes annotation quality over dataset size, ensuring that our benchmark measurements reflect genuine model capabilities rather than label noise.

**Quality Control Measures.** Beyond the dual-phase workflow, we implement additional quality controls: (1) *Selective task inclusion*: We exclude tasks that exceed typical annotator capabilities or present inherently ambiguous correctness criteria, noting that even human performance on OSWorld reaches only 72.36%. (2) *Orthogonal annotation*: We deliberately annotate bad actions within successful trajectories and good actions within failed trajectories, recognizing that trajectory success and step correctness are independent dimensions. This enables robust evaluation across varying trajectory contexts and prevents models from simply learning trajectory-level correlations. Rather than retaining disputed examples and only reporting inter-annotator agreement, our protocol actively resolves disagreements through mandatory consensus review and excludes cases that remain ambiguous. Although we did not record the initial disagreement rate, the final benchmark contains only high-confidence annotations that passed independent cross-validation and consensus resolution.

### A.3. VLM Selection As Reward Models

Considering the practical application scenarios of CUARewardBench—large-scale data construction and online reinforcement learning training—we primarily evaluate open-source models while also including representative frontier proprietary models to contextualize current capability limits. This choice reduces the implementation difficulty for reproducing our research, considers the feasibility of large-scale deployment in real-world applications, and provides strong closed-source reference points. We evaluate four categories of models (12 models in total):

- *General VLMs*: We assess the Qwen2.5VL series (Bai et al., 2025b), which represents the leading open-source vision-language models. We evaluate three variants within this series: 7B, 32B, and 72B parameters.

- *Visual Reasoning Models*: We include GLM-4.5V-106B (Hong et al., 2025), a mixture-of-experts model with 12B activated parameters, Qwen3VL-32B-Thinking and Qwen3VL-235B-Thinking (Bai et al., 2025a), which are advanced reasoning-enhanced vision-language models, and representative proprietary frontier models, including GPT-5 (OpenAI, 2025a), Qwen3.5-397B-A17B-FP8, and Kimi-K2.5. We report the official model names available at the time of evaluation for these proprietary systems.

- *Specialized CUA Models*: CUA models are endowed with extensive CUA-related knowledge, intuitively suggesting they should possess CUA trajectory evaluation capabilities. However, most CUA models fail to follow instructions for trajectory evaluation, which we attribute to catastrophic forgetting caused by extensive CUA data during post-training. GUI-OWL (Ye et al., 2025) series represents an exception, built upon Qwen2.5VL with post-training data mixing CUA training and general reasoning data. This approach enables strong CUA capabilities (GUI-OWL-7B achieves 29.4% on OSWorld) while maintaining robust reasoning abilities. We evaluate both GUI-OWL-7B and 32B as reward models.

- *Specialized CUA Reward Models*: World State Model of SE-Agent (Sun et al., 2025) (SE-WSM) is the only existing open-source CUA-specialized reward model, based on Qwen2.5VL-7B with dedicated CUA reward training. Its training data comprises 860 trajectories from 43 feasible Chrome tasks in OSWorld, executed by UI-TARS and Gemini-2.5-Pro, and automatically annotated by GPT-4o.

### A.4. Prompt Templates of Reward Models

For ZeroGUI (Yang et al., 2025a) and SE-WSM (Sun et al., 2025) prompts, we directly adopt the original versions from their open-source implementations, as shown in Figure 3 and Figure 4. For the OpenCUA reflector (Wang et al., 2025b) prompt, we modified the components coupled with step-wise chain-of-thought reasoning to ensure compatibility with our experimental environment. The original and simplified prompts are presented in Figure 5 and Figure 6, respectively.

### A.5. Supplementary Results for Additional Task Categories

This section presents the experimental results for the remaining 5 task categories that were not included in the main paper due to space constraints. These results complement the overall performance and 5 selected categories (VS Code, GIMP, LibreOffice Writer, Chrome, and Multi-apps) shown in Tables 4 and 5, providing complete coverage of all 10 software categories in CUARewardBench.

---

**ORM Prompt Template of ZeroGUI**

You are an expert at analyzing computer usage task completion from screenshots.

You will be given a task instruction and a series of screenshots of the task execution. Please analyze the screenshots and provide a detailed analysis of the task completion by following the steps below:

1. First, analyze and understand the task instruction. Describe what should the screenshots look like if the task is completed successfully.

2. Describe what you observe in each screenshot, analysis what actions were taken and what changes were made to the UI to achieve the task (or mistakes made).

3. When you analyze the screenshots, please pay attention to the very detailed elements and changes in the UI. Every small detail may affect the final result.

4. After all screenshots are analyzed, provide a overall reasoning about how the task was completed or failed at **the final state**. Make sure you have considered all demands of the task instruction.

5. Determine if the task was completed at **the final state** (the last screenshot) successfully (score 1 for success, 0 for failure). If the task is completed during the process but not at the final state, it should be considered as failure (0 score). Provide your response strictly in the following format:

TASK REQUIREMENT:

[Your understanding of the task instruction]

SCREENSHOT ANALYSIS:

Screenshot 1:

[Analysis of first screenshot]

Screenshot 2:

[Analysis of second screenshot]

...

REASONING:

[Your reasoning]

FINAL ANSWER:

[Your final answer]

SCORE: [0/1]

Now, please **strictly follow the format** and analyze the following screenshots (The last line should only be SCORE: [0/1], no other text):

Task Instruction: {instruction}

Screenshots (by order):

*Figure 3.* ORM prompt template of ZeroGUI (Yang et al., 2025a). The prompt instructs the vision-language model to analyze computer usage task completion through detailed screenshot examination and structured response formatting.

---

**ORM and PRM Prompt Template of SE-WSM**

You are an expert at analyzing computer usage task completion from screenshots. I am evaluating the performance of a UI agent. The images provided are **sequential keyframes** that represent the full execution trajectory of the agent when attempting to follow a command. These keyframes correspond to the instruction: **'{instruction}'**.

Please thoroughly analyze the sequence to assess the following aspects:

1. **Correctness** — Did the agent successfully complete the task as instructed?
2. **Redundant Steps** — Identify any unnecessary or repeated actions that do not contribute to the goal.
3. **Optimization** — Did the agent follow an efficient plan with a minimal number of steps?
4. **First Error Step** — If the execution is incorrect or sub-optimal, determine the index of the **first keyframe where a mistake occurred**.
5. **Error Analysis** — Provide a brief explanation of the mistake at that step.
6. **Correct Action Suggestion** — Explain what the agent **should have done instead** at the point of error.

**Important Instructions:**

- The agent may have made progress toward the goal, but unless the task is **fully and correctly completed**, you must set 'Correctness' to **False**.
- Be cautious in determining success. Missing confirmation screens, skipped inputs, or wrong UI elements clicked all count as errors.
- Carefully examine all UI changes, button interactions, text entries, and any visual feedback in the screenshots.
- Clearly indicate **which exact steps are redundant** (starting from 1).

Once you finish the analysis, return your evaluation in the following dictionary format (include your step-by-step reasoning **above** the result):

<analysis process>
your step-by-step reasoning
</analysis process>
<res_dict>
{
"Correctness": True/False,
"Redundant": [step_num, ...],
"Optimized": True/False,
"First_Error_Step": step_num or None,
"Error_Type": "brief description of the mistake",
"Correct_Action": "what should have been done instead"
}
</res_dict>

*Figure 4.* SE-WSM (Sun et al., 2025) prompt template for both ORM and PRM evaluation. The prompt instructs the vision-language model to conduct multi-dimensional assessment including trajectory correctness, redundant steps identification, first error step detection, and correct action suggestions.

---

**PRM Prompt Template of OpenCUA Reflector**

You are a judge of a computer-use agent. You will be given a task, the agent's history actions, agent last action and thought process with 2 screenshots.
- Thought is the reasoning for the history steps and prediction for the next step. - Action is the summary of the code - Code is the code that will be executed. - The first screenshot is the observation of the last action and the second image is the computer state after executing the last action (code).
**Task:** {goal}
**History steps:** {history_steps}
**Last step:**
**Thought:** {thought}
**Action:** {action}
**Code:** {code}
Your response should include 3 parts:
**1. Is the last step redundant:** - If the last step is doing unnecessary action or action that is not related to the task, for example, clicking irrelevant places, open irrelevant applications, or unnecessary scrolls, you should mark it as redundant.
**2. Is the last step incorrect:** - If the action is related to the task but executing the code did not produce the expected change, you should mark it as incorrect. - If the action and the code do not align, you should mark it as incorrect. For example the action tries to click an element but failed according to the screenshot. - The last screenshot shows the application or window is not fully loaded, but the code is executed. - If there is any mistake in the thought action.
**3. Reflection:** - You should first provide a natural summary of the visual changes between the last screenshot and the current screenshot. If there is no change, please mention it. - If the last step is correct and not redundant, you should then say the step is necessary and how it is effective. - If the last step is incorrect, you should then provide a clear explanation of the error. - If the last step is redundant, you should then provide a clear explanation.
**YOUR RESPONSE MUST BE EXACTLY ONE VALID JSON OBJECT. NO MARKDOWN, NO EXTRA TEXT.**
Here is the exact JSON structure you must follow:
<res_dict>
{
"last_step_correct": bool,
"last_step_redundant": bool,
"reflection": str
}
</res_dict>

*Figure 5.* Original PRM prompt template of OpenCUA reflector (Wang et al., 2025b). The prompt provides comprehensive step-level assessment including thought process analysis, action-code alignment verification, and structured JSON output format for systematic evaluation of agent decision-making processes.

---

**Simplified PRM Prompt Template of OpenCUA Reflector**

You are a judge of a computer-use agent. Your role is to evaluate whether the agent's last action was redundant, incorrect, or appropriate for completing the given task. You will analyze screenshots showing the agent's history, the state before the last action, and the state after the last action, along with the raw code that was executed.

You will be given a task, the agent's history screenshots, agent's last action code, and 2 screenshots showing before and after the last action.

- The history screenshots show what happened before the last step, helping you understand the agent's previous progress and context. - The last action is represented by the raw code that was executed. - The before and after screenshots show the state immediately before and after executing the last action code. - Note: If there is mouse related code that needs coordinates, the center of the red circle in the before screenshot shows the position. But do not mention the red circle or red dot in any part of your response. - You will be provided with: history screenshots (showing previous steps), one screenshot before executing the last action, and one screenshot after executing the last action. - You should focus on the differences between the before and after screenshots to understand what the last action accomplished, while using history screenshots to understand the context and detect redundancy.

**Task:** {instruction}
**History screenshots:** <image>
**Screenshots before and after the last action:** <image>
**Last action code:** Step {step_index}: {action_code}
Your response should include 3 parts:
**1. Is the last step redundant:** - If the last step is doing unnecessary action or action that is not related to the task, for example, clicking irrelevant places, open irrelevant applications, or unnecessary scrolls, you should mark it as redundant. - If the last step is a repeat of a former step based on the history screenshots, you should mark it as redundant. - Too many scrolls or drags of the scroll bar, or too many clicks of the same button, or too many clicks of the same element, you should mark it as redundant.
**2. Is the last step incorrect:** - If the action is related to the task but executing the code did not produce the expected change, you should mark it as incorrect. - If the code execution failed or did not work as intended based on the before/after screenshots, you should mark it as incorrect. - The after screenshot shows the application or window is not fully loaded, but the code was executed. - If there is any clear mistake in the action based on what the code was trying to accomplish. - You should carefully examine the click/drag related actions. In many cases, the code wants to click a target, but it doesn't match the element at the center of the red circle in the before screenshot.
**3. Reflection:** - You should first provide a natural summary of the visual changes between the before and after screenshots. If there is no change, please mention it. - If the last step is correct and not redundant, you should then say the step is necessary and how it is effective toward completing the task. - If the last step is incorrect, you should then provide a clear explanation of the error. - If the last step is redundant, you should then provide a clear explanation of why it's unnecessary given the history.
Once you finish the analysis, return your evaluation in the following dictionary format (include your step-by-step Reflection **above** the result):
<analysis process>
[your step-by-step reflection]
</analysis process>
<res_dict>
{
"last_step_correct": bool,
"last_step_redundant": bool,
"reflection": str
}
</res_dict>

---

*Figure 6.* Simplified PRM prompt template of OpenCUA reflector (Wang et al., 2025b). The prompt instructs the vision-language model to assess individual agent actions by analyzing visual changes between consecutive screenshots and determining whether the latest action is correct, redundant, or necessary for task completion.

| Reward Model | Overall | | | | | vlc | | os | | thunderbird | | impress | | calc | |
|---|---|---|---|---|---|---|---|---|---|---|---|---|---|---|---|
| | P | NPV | R | S | OA | P | NPV | P | NPV | P | NPV | P | NPV | P | NPV |
| *zerogui* | | | | | | | | | | | | | | | |
| Qwen2.5VL-7B | 60.8 | 65.0 | 74.8 | 49.2 | 62.1 | 71.4 | 66.7 | 61.1 | 62.5 | 50.0 | 50.0 | 47.6 | 42.9 | 65.0 | 71.4 |
| Qwen2.5VL-32B | 70.9 | 76.1 | 80.6 | 65.2 | 72.8 | 72.7 | 55.6 | 64.3 | 58.3 | **100.0** | 80.0 | 66.7 | 77.8 | 68.2 | 83.3 |
| Qwen2.5VL-72B | 70.0 | 72.1 | 75.5 | 66.2 | 71.0 | 83.3 | 50.0 | 66.7 | 63.6 | **100.0** | 72.7 | 63.2 | 77.8 | 63.2 | 66.7 |
| GLM-4.5V-106B | 76.8 | 90.4 | **92.8** | 70.7 | 82.0 | 91.7 | **87.5** | 68.4 | 85.7 | **100.0** | 80.0 | 70.6 | 81.8 | 78.9 | 86.7 |
| Qwen3VL-32B-T | 80.6 | **91.1** | **92.8** | 76.7 | 84.9 | 84.6 | 85.7 | 72.2 | 87.5 | 88.9 | **100.0** | 77.8 | **100.0** | 78.9 | 86.7 |
| Qwen3VL-235B-T | 82.6 | 90.6 | 92.1 | 79.7 | 86.0 | 90.9 | 77.8 | **73.7** | **100.0** | **100.0** | 88.9 | 75.0 | 83.3 | 85.0 | **100.0** |
| GPT-5 | **86.3** | 89.7 | 90.6 | **85.0** | **87.9** | 100.0 | 72.7 | 72.2 | 87.5 | **100.0** | 88.9 | **81.2** | 91.7 | 100.0 | 94.4 |
| GUI-OWL-7B | 69.0 | 68.8 | 71.9 | 65.6 | 68.4 | 75.0 | 50.0 | 71.4 | 66.7 | 80.0 | 63.6 | 58.3 | 56.2 | 62.5 | 61.1 |
| GUI-OWL-32B | 69.0 | 73.7 | 78.4 | 63.2 | 71.0 | 100.0 | 61.5 | 68.8 | 70.0 | 75.0 | 75.0 | 50.0 | 50.0 | 59.1 | 66.7 |
| *sewsm* | | | | | | | | | | | | | | | |
| Qwen2.5VL-7B | 63.1 | 57.1 | 50.4 | 69.2 | 59.6 | 50.0 | 35.7 | 75.0 | 55.6 | 60.0 | 54.5 | 66.7 | 62.5 | 55.6 | 56.2 |
| Qwen2.5VL-32B | 80.0 | 71.7 | 69.1 | 82.0 | 75.4 | 75.0 | 50.0 | 81.8 | 66.7 | **100.0** | 72.7 | 85.7 | 85.7 | 78.6 | 70.0 |
| Qwen2.5VL-72B | 78.6 | 74.5 | 74.1 | 78.9 | 76.5 | 77.8 | 54.5 | 73.3 | 72.7 | 66.7 | 60.0 | 75.0 | 68.8 | 82.4 | 82.4 |
| GLM-4.5V-106B | 82.9 | 77.6 | 77.0 | 83.5 | 80.1 | 100.0 | 53.3 | **86.7** | **90.9** | **100.0** | 72.7 | 75.0 | 83.3 | 91.7 | 72.7 |
| Qwen3VL-32B-T | 85.3 | 83.1 | 83.5 | 85.0 | 84.2 | 87.5 | 58.3 | 75.0 | 80.0 | 87.5 | 87.5 | 85.7 | 85.7 | **100.0** | **94.4** |
| Qwen3VL-235B-T | 85.7 | 73.6 | 77.7 | 86.4 | 81.6 | **100.0** | 72.7 | 76.5 | 87.5 | **100.0** | 80.0 | 83.3 | 75.0 | 92.3 | 76.2 |
| GPT-5 | 89.0 | 86.6 | 87.1 | 88.5 | 87.1 | **100.0** | 57.1 | 81.2 | 90.0 | **100.0** | 87.5 | 75.0 | 81.8 | 100.0 | 89.5 |
| Qwen3.5-397B-A17B-FP8 | **89.3** | 84.4 | 84.2 | 89.5 | 86.8 | **100.0** | 47.1 | 81.2 | 90.0 | **100.0** | **100.0** | 92.9 | 92.9 | 93.8 | 88.9 |
| Kimi-K2.5 | 88.4 | **92.1** | **92.8** | 87.2 | **90.1** | **100.0** | **88.9** | 72.2 | 87.5 | **100.0** | **100.0** | 86.7 | 92.3 | 100.0 | 85.0 |
| GUI-OWL-7B | 68.4 | 72.4 | 76.8 | 63.2 | 69.9 | 87.5 | 58.3 | 64.7 | 75.0 | **100.0** | 80.0 | 50.0 | 50.0 | 63.6 | 75.0 |
| GUI-OWL-32B | 75.0 | 71.4 | 71.2 | 75.2 | 73.2 | 85.7 | 57.1 | 73.3 | 72.7 | 66.7 | 71.4 | 62.5 | 66.7 | 72.7 | 60.9 |
| SE-WSM-7B | 70.0 | 52.2 | 20.1 | **91.0** | 54.8 | **100.0** | 57.1 | 33.3 | 40.0 | 0.0 | 50.0 | 0.0 | 48.1 | 33.3 | 48.4 |
| *voting-majority* | | | | | | | | | | | | | | | |
| G106-s 2runs | 84.3 | 70.7 | 65.5 | 87.2 | 76.1 | **100.0** | 53.3 | 83.3 | 71.4 | **100.0** | 66.7 | 83.3 | 75.0 | 90.0 | 66.7 |
| Q32-s + G106-s | 90.1 | 68.5 | 59.0 | **93.2** | 75.7 | **100.0** | 50.0 | 90.0 | 68.8 | **100.0** | 66.7 | **100.0** | **87.5** | 87.5 | 61.5 |
| Q32-s + G106-s + G106-z | 81.6 | **84.8** | **86.3** | 79.7 | 83.1 | **100.0** | **61.5** | 85.7 | **83.3** | **100.0** | 80.0 | 75.0 | 83.3 | 93.8 | 88.9 |
| Q3-32-s + Q3-235-z | 88.0 | 80.3 | 79.1 | 88.7 | **83.8** | **100.0** | **61.5** | 75.0 | 80.0 | **100.0** | 80.0 | 91.7 | 81.2 | **100.0** | **94.4** |
| *voting-strict_unanimous* | | | | | | | | | | | | | | | |
| G106-s 2runs | 84.3 | 82.3 | 65.5 | **76.7** | 71.0 | **100.0** | 53.8 | 90.0 | 90.0 | **100.0** | 80.0 | 83.3 | 85.7 | 90.0 | 82.4 |
| Q32-s + G106-s | 90.1 | 84.2 | 59.0 | 72.2 | 65.4 | **100.0** | 54.5 | 90.0 | 90.0 | **100.0** | 80.0 | **100.0** | 80.0 | 87.5 | 87.5 |
| Q32-s + G106-s + G106-z (**w/ UPE**) | 89.8 | 93.3 | 56.8 | 63.2 | 59.9 | **100.0** | **83.3** | 90.0 | **100.0** | **100.0** | 80.0 | **100.0** | 87.5 | 85.7 | **91.7** |
| Q3-32-s + Q3-235-z (**w/ UPE**) | 88.0 | **95.3** | **79.1** | 75.9 | **77.6** | **100.0** | 75.0 | 75.0 | **100.0** | **100.0** | 80.0 | **100.0** | **100.0** | 91.7 | 90.0 |

*Table 7.* Supplementary results for Table 4, showing performance of outcome reward models (ORM) on the remaining task categories (vlc, os, thunderbird, impress, calc). Results show precision (P) and negative predictive value (NPV) for trajectory success evaluation under multiple prompt configurations: *zerogui*, *sewsm*, *voting-majority*, and *voting-strict_unanimous*. In model names, "-z" denotes *zerogui* prompt and "-s" denotes *sewsm* prompt; "Q32" denotes Qwen2.5VL-32B, "G106" denotes GLM-4.5V-106B, "Q3-32" denotes Qwen3VL-32B, and "Q3-235" denotes Qwen3VL-235B. The Overall metrics remain the same as in Table 4.

The supplementary categories include LibreOffice Calc, LibreOffice Impress, VLC, Thunderbird, and OS operations. T hese results maintain consistency with the patterns observed in the main text, further validating our key findings.

### A.6. End-to-End Reward Model Validation

Beyond diagnostic metrics, **we provide direct empirical evidence that benchmark-guided reward models improve downstream agent performance**. To validate that our benchmark measurements translate to practical utility, we conduct a supervised fine-tuning (SFT) experiment where a strong open-source reward model identified by CUARewardBench is used to filter training data for agent improvement.

**Experimental Setup.** We design a controlled end-to-end validation experiment in the LibreOffice Impress domain:

1. **Task Construction**: We construct 437 tasks within the LibreOffice Impress domain, covering diverse presentation creation and editing operations.
2. **Trajectory Generation**: Using Qwen3VL-32B-Thinking as the policy model, we generate 10 rollout attempts per task, yielding 4,370 trajectories in total.
3. **Reward-Based Filtering**: We employ Qwen3VL-32B-Thinking (a strong open-source model identified by our bench-mark) as the reward model to identify successful trajectories. Although this reward model uses the same base VLM

| Reward Model | Overall | | | | | vlc | | os | | thunderbird | | impress | | calc | |
|---|---|---|---|---|---|---|---|---|---|---|---|---|---|---|---|
| | P | NPV | R | S | OA | P | NPV | P | NPV | P | NPV | P | NPV | P | NPV |
| *opencua_reflector* | | | | | | | | | | | | | | | |
| Qwen2.5VL-7B | 54.4 | 49.4 | 53.8 | 50.0 | 52.0 | 62.5 | 40.0 | 47.1 | 54.5 | 30.0 | 44.4 | 71.4 | 54.5 | 55.6 | 47.4 |
| Qwen2.5VL-32B | 60.3 | 64.8 | 79.8 | 41.5 | 61.7 | 68.4 | 57.1 | 55.0 | 66.7 | 42.9 | 60.0 | 71.4 | 66.7 | 50.0 | 38.5 |
| Qwen2.5VL-72B | 58.5 | 64.8 | 83.0 | 34.8 | 60.1 | 75.0 | 83.3 | 45.5 | 50.0 | 41.7 | 57.1 | 60.0 | 50.0 | 58.8 | 62.5 |
| GLM-4.5V-106B | 64.0 | 78.5 | 89.0 | 44.5 | 67.9 | 72.7 | **100.0** | 47.8 | 60.0 | 50.0 | 80.0 | 72.2 | 61.1 | 64.5 | **100.0** |
| Qwen3VL-32B-T | 68.3 | 77.3 | 85.2 | 56.1 | 71.4 | **92.3** | 69.2 | 47.8 | 60.0 | 66.7 | **100.0** | 65.2 | 61.5 | 60.7 | 66.7 |
| Qwen3VL-235B-T | 66.5 | 75.0 | 84.1 | 53.0 | 69.4 | 76.5 | 66.7 | 54.2 | **100.0** | 57.1 | **100.0** | 60.0 | 54.5 | **66.7** | 80.0 |
| GPT-5 | **72.7** | **87.8** | **92.3** | **61.6** | **77.7** | 88.2 | 88.9 | **60.0** | 87.5 | 66.7 | **100.0** | 67.9 | **87.5** | 65.5 | 87.5 |
| GUI-OWL-7B | 64.6 | 61.8 | 67.0 | 59.1 | 63.3 | 76.5 | 66.7 | 55.6 | 70.0 | **71.4** | 75.0 | **76.5** | 63.2 | 61.9 | 56.2 |
| GUI-OWL-32B | 61.6 | 61.5 | 71.4 | 50.6 | 61.6 | 70.0 | 66.7 | 55.0 | 75.0 | 42.9 | 58.3 | 61.9 | 53.3 | 57.9 | 50.0 |
| *sewsm* | | | | | | | | | | | | | | | |
| Qwen2.5VL-7B | 56.7 | 54.2 | 67.0 | 43.3 | 55.8 | 73.7 | 71.4 | 42.1 | 44.4 | 33.3 | 42.9 | 75.0 | **68.8** | 51.7 | 37.5 |
| Qwen2.5VL-32B | 65.3 | 62.4 | 67.8 | 59.8 | 64.0 | 76.5 | 66.7 | 53.3 | 57.1 | 55.6 | 70.0 | 58.3 | 50.0 | 47.8 | 35.7 |
| Qwen2.5VL-72B | 58.7 | 67.1 | 85.2 | 33.5 | 60.7 | 72.7 | **100.0** | 45.0 | 50.0 | 50.0 | 76.9 | 60.0 | 54.5 | 54.8 | 50.0 |
| GLM-4.5V-106B | 69.5 | 64.2 | 66.1 | 67.7 | 66.9 | 76.5 | 66.7 | 53.8 | 56.2 | 57.1 | 66.7 | 61.9 | 53.3 | 73.3 | 59.1 |
| Qwen3VL-32B-T | 78.2 | 66.3 | 63.2 | 80.5 | 71.4 | 90.9 | 60.0 | **75.0** | **75.0** | 83.3 | 76.9 | 63.2 | 52.9 | **85.7** | 65.2 |
| Qwen3VL-235B-T | 74.5 | 63.2 | 59.3 | 77.4 | 67.9 | **100.0** | 66.7 | 69.2 | 73.3 | 62.5 | 72.7 | 66.7 | 52.4 | 66.7 | 54.5 |
| GPT-5 | **83.7** | 68.8 | **89.0** | **86.0** | **74.9** | **100.0** | 71.4 | 69.2 | 73.3 | 71.4 | 75.0 | 68.2 | 64.3 | 84.6 | 62.5 |
| Qwen3.5-397B -A17B-FP8 | 77.0 | 63.8 | 58.8 | 80.5 | 69.1 | 78.6 | 58.3 | 57.1 | 64.3 | **100.0** | 78.6 | 63.6 | 57.1 | 76.9 | 58.3 |
| Kimi-K2.5 | 72.4 | 65.0 | 64.8 | 72.6 | 68.5 | 85.7 | 66.7 | 43.8 | 50.0 | 57.1 | 66.7 | **68.4** | 58.8 | 77.8 | **68.4** |
| GUI-OWL-7B | 56.6 | 64.2 | 86.8 | 26.2 | 58.1 | 66.7 | 60.0 | 48.0 | 66.7 | 70.0 | **88.9** | 58.6 | 57.1 | 56.7 | 57.1 |
| GUI-OWL-32B | 57.4 | **68.8** | 86.8 | 26.8 | 59.5 | 66.7 | 60.0 | 50.0 | 66.7 | 46.7 | 75.0 | 57.6 | 66.7 | 53.8 | 45.5 |
| SE-WSM-7B | 58.7 | 52.0 | 48.4 | 62.2 | 54.9 | 75.0 | 60.0 | 30.8 | 40.0 | 33.3 | 50.0 | 56.2 | 45.0 | 50.0 | 43.5 |
| *voting-majority* | | | | | | | | | | | | | | | |
| G106-s 2runs | 73.6 | 62.8 | 59.6 | 76.2 | 67.4 | 86.7 | 72.7 | 60.0 | 57.9 | 66.7 | 62.5 | 61.9 | 53.3 | 76.9 | 58.3 |
| Q32-s + G106-s | 75.6 | 60.5 | 52.5 | 81.1 | 66.0 | 85.7 | 66.7 | 55.6 | 55.0 | 57.1 | 66.7 | 58.8 | 47.4 | 66.7 | 50.0 |
| Q32-s + G106-s + G106-o | 68.4 | **71.5** | 78.6 | 59.8 | 69.7 | 77.8 | **75.0** | 47.1 | 54.5 | 50.0 | 63.6 | **62.5** | **58.3** | 59.3 | 60.0 |
| Q3-32-s + Q3-235-o | **83.1** | 63.2 | 53.8 | **87.8** | 69.9 | **100.0** | 58.8 | **75.0** | **75.0** | **83.3** | **76.9** | **62.5** | 50.0 | **84.6** | **62.5** |
| *voting-strict_unanimous* | | | | | | | | | | | | | | | |
| G106-s 2runs | 73.6 | 64.2 | **59.6** | **57.9** | **58.8** | 86.7 | 57.1 | 60.0 | 50.0 | 66.7 | 63.6 | 61.9 | 54.5 | 76.9 | 64.7 |
| Q32-s + G106-s | 75.6 | 69.1 | 52.5 | 46.3 | 49.6 | 85.7 | 66.7 | 55.6 | 60.0 | 57.1 | 70.0 | 58.8 | 62.5 | 66.7 | 50.0 |
| Q32-s + G106-s + G106-o (**w/ UPE**) | 81.7 | 85.1 | 48.9 | 24.4 | 37.3 | 92.3 | **100.0** | 55.6 | **100.0** | 57.1 | **100.0** | 80.0 | 57.1 | 75.0 | **100.0** |
| Q3-32-s + Q3-235-o (**w/ UPE**) | **83.1** | **86.2** | 53.8 | 45.7 | 50.0 | **100.0** | 71.4 | **75.0** | **100.0** | **83.3** | **100.0** | 62.5 | **62.5** | **84.6** | 88.9 |

*Table 8.* Supplementary results for Table 5, showing performance of process reward models (PRM) on the remaining task categories (vlc, os, thunderbird, impress, calc). Results show precision (P) and negative predictive value (NPV) for step-level correctness assessment under multiple prompt configurations: *opencua_reflector*, *sewsm*, *voting-majority*, and *voting-strict_unanimous*. In model names, "-o" denotes *opencua_reflector* prompt and "-s" denotes *sewsm* prompt; "Q32" denotes Qwen2.5VL-32B, "G106" denotes GLM-4.5V-106B, "Q3-32" denotes Qwen3VL-32B, and "Q3-235" denotes Qwen3VL-235B. The Overall metrics remain the same as in Table 5.

family as the rollout policy, it is applied with an independent reward-evaluation prompt and does not access OSWorld evaluation labels. The reward model successfully verifies 2,243 trajectories as correct. To ensure training data quality and balance task difficulty, we apply per-task filtering to exclude overly simple tasks (those with excessive successful rollouts) and retain a curated set of 1,532 high-quality trajectories for supervised fine-tuning.

4. **Agent Training**: We fine-tune Qwen3VL-7B-Thinking on the filtered 1,532 trajectories using standard supervised learning.

5. **Evaluation**: We evaluate both the pre-trained baseline and the fine-tuned agent on the original 47 LibreOffice Impress tasks from OSWorld, conducting 10 independent trials per task to measure task-level success rates.

**Results.** The agent fine-tuned on reward-model-filtered trajectories achieves a task success rate of 46.10%, compared to 34.68% for the baseline pre-trained model—representing a **33% relative improvement**. This substantial performance gain demonstrates that our benchmark metrics meaningfully predict real-world utility: reward models that perform well on CUARewardBench can identify high-quality training data that translates to improved agent capabilities. This end-to-end validation provides stronger evidence than statistical correlation alone, as it directly answers whether benchmark performance translates to practical agent improvement in downstream applications.

## A.7. Evaluation Robustness

CUARewardBench is designed as a fixed offline classification benchmark rather than an online environment benchmark with stochastic rollouts. Each reward model evaluates the same static set of trajectories, screenshots, task instructions, and key-step annotations under predetermined prompt templates. Therefore, the reported metrics directly measure reward classification reliability on a fixed benchmark set, and repeated environmental trials are unnecessary for the benchmark evaluation itself.

While VLM inference can exhibit minor sampling variance, this variance is substantially smaller than the environmental randomness typically observed in online CUA evaluation. We further reduce avoidable variance by using fixed evaluation inputs and consistent parsing rules across models. The resulting performance patterns are coherent across model families, prompt templates, ORM/PRM settings, and ensemble configurations; for example, stronger reasoning models consistently improve reliability, PRM remains more challenging than ORM, and UPE systematically increases precision and NPV by abstaining on non-unanimous samples. These consistent patterns support the robustness of the benchmark conclusions.

## A.8. Error Analysis

This section provides a comprehensive microscopic analysis of error patterns, examining specific failure modes, and the fundamental limitations that constrain current reward model effectiveness. We analyze 53 failure cases from GLM-4.5V-106B, a representative strong ORM model in our evaluation, to identify systematic error patterns. As shown in Table 9, we categorize errors by frequency: reasoning errors (35.8%), visual understanding errors (30.2%), action understanding errors (17.0%), knowledge deficiency (15.1%), and inherent RM limitations (1.9%). We examine each category in detail below.

| Error Category | Count | Percentage (%) |
|---|---|---|
| Reasoning Error | 19 | 35.8 |
| Visual Understanding Error | 16 | 30.2 |
| Action Understanding Error | 9 | 17.0 |
| Knowledge Deficiency | 8 | 15.1 |
| Inherent RM Limitation | 1 | 1.9 |
| **Total** | **53** | **100** |

*Table 9.* Distribution of error modes of GLM-4.5V-106B as ORM using *sewsm* prompt.

**Visual Understanding Errors (30.2%).** Reward models frequently misinterpret visual information in screenshots, leading to incorrect assessments of computer states. For instance, when an agent executes the task "adding strike-through sign on the line," it successfully selects and applies strike-through formatting but misses the final few characters. The reward model fails to detect this incomplete execution and incorrectly judges the trajectory as successful.

**Action Understanding Errors (17.0%).** Under the SE-WSM prompt configuration, reward models infer agent actions from consecutive screenshots to evaluate trajectory success. However, models often derive incorrect actions from adjacent frames. For example, when an agent clicks an "OK" button in a dialog box, the reward model mistakenly believes the agent failed to complete the confirmation operation, leading to an incorrect failure assessment. A straightforward solution involves incorporating coordinate markers in screenshots, as implemented in the OpenCUA reflector approach, which significantly reduces such errors.

**Knowledge Deficiency (15.1%).** Just as agents frequently fail due to insufficient software operation knowledge, reward models often lack domain-specific knowledge necessary to establish correct task success criteria. For instance, when an agent's task is to "enlarge the text on my screen", the agent incorrectly magnifies the entire screen rather than adjusting text font size. The reward model, unaware that Ubuntu system settings provide distinct options for these operations, incorrectly judges the trajectory as successful.

**Reasoning Errors (35.8%).** Even when reward models correctly understand visual elements and agent actions while possessing relevant knowledge, they frequently commit logical errors during information synthesis. For example, when an agent successfully completes the task "set the decimal separator as a comma (,)", the reward model initially acknowledges the correct configuration but subsequently engages in convoluted reasoning that leads to overturning its original correct conclusion.

**Inherent RM Limitations (1.9%).** A small but significant category of errors reveals fundamental limitations of VLM-based

reward models: screenshots provide only partial observations of computer states. For instance, when an agent successfully completes the task "use GIMP to compress the image to under 600KB", the screenshot lacks visual feedback about the compressed file size, leaving the reward model without evidence to verify task completion. This limitation suggests that reward models and script-based verifiers could serve as complementary approaches for more robust reward estimation.

### A.9. Future Directions

While our work establishes a comprehensive benchmark and proposes an effective ensemble method for CUA reward models, our findings reveal several promising directions for future research.

**Advancing Process Reward Models.** Our evaluation reveals that process reward models (PRMs) face significantly greater challenges than outcome reward models (ORMs). Across the evaluated single VLMs, PRM performance remains substantially lower than ORM performance, particularly in terms of jointly maintaining high precision and NPV. This performance gap stems from the inherent difficulty of step-level evaluation, which requires fine-grained understanding of action semantics and their effects on GUI states. The action understanding errors (17.0%) identified in our error analysis particularly highlight this challenge. Future work should prioritize developing specialized architectures and training strategies tailored for step-level assessment.

**Enhancing Visual Reasoning and Understanding Capabilities on GUI.** Our error analysis reveals that visual reasoning errors (35.8%) and visual understanding errors (30.2%) together constitute 66% of all failure cases, establishing GUI-specific visual reasoning as the core bottleneck for CUA reward models. Current VLMs struggle with fine-grained GUI element recognition, spatial relationship understanding in complex layouts, and multi-step logical inference over GUI state transitions. Future research should focus on developing specialized training data and methodologies for reward models, including: (1) curating high-quality GUI-specific verification datasets with diverse failure modes and edge cases, (2) designing reward model training objectives that explicitly emphasize visual grounding and step-level reasoning, and (3) exploring curriculum learning strategies that progressively expose models to increasingly complex GUI verification tasks, from simple single-action validations to multi-step workflow assessments.

**Exploring Hybrid Verification Approaches.** Our analysis identifies inherent reward model limitations (1.9% of errors) stemming from the fundamental constraint that screenshots provide only partial observations of computer states—information invisible in visual frames, such as clipboard contents, background processes, or system configurations, remains inaccessible to vision-based models. This limitation suggests that purely vision-based reward models may face a theoretical ceiling in verification accuracy. Future work should investigate hybrid approaches that strategically combine VLM-based reward models with complementary verification methods, such as LLM-based reasoning over structured state representations (e.g., accessibility trees, DOM structures) and script-based verifiers for deterministic state checks. Such hybrid systems could leverage the strengths of each component: VLMs for visual understanding, LLMs for logical reasoning over symbolic representations, and script-based verifiers for precise state validation, potentially achieving more comprehensive and reliable trajectory verification than any single approach alone.

**Improving Deployment Efficiency.** Although our study focuses on reward model reliability, practical CUA training pipelines may require large-scale reward inference over massive trajectory collections. Future work could therefore explore efficient deployment techniques for CUA reward models, including post-training quantization, mixed-precision inference, and network pruning. These approaches are complementary to our reliability-oriented benchmark and may reduce the computational cost of applying reward models at scale.

### A.10. Use of Large Language Models

We disclose that Large Language Models (LLMs) were used in the preparation of this paper. Specifically, LLMs were used for writing assistance, code/debugging assistance, and editorial refinement, including improving clarity, grammar, and overall presentation of the content. All technical decisions, experiments, annotations, analyses, and claims were reviewed and verified by the authors.

