# OpenReview forum: "CUARewardBench: A Benchmark for Evaluating Reward Models for Computer-Using Agents"
_ICML.cc/2026/Conference — ICML 2026 regular_

### Official Review · Reviewer_Y7JD · 2026-02-21

**Soundness:** 3
**Presentation:** 3
**Significance:** 3
**Originality:** 3
**Overall Recommendation:** 4
**Confidence:** 4

**Summary:**

This paper proposes the first comprehensive benchmark, CUARewardBench, to address the issue of the lack of standardized reward model evaluation tools for computer proxy usage. This benchmark includes a high-quality and diverse trajectory dataset, which is used to systematically evaluate multiple visual language models. Research has found that general VLM is superior to specialized CUA models in evaluating tasks, and visual reasoning ability is key. Based on this, the author proposes a novel "consistent prompt ensemble" method that can significantly improve the accuracy and negative predictive value of the reward model, providing a practical solution for building a more reliable CUA training process.

**Compliance With Llm Reviewing Policy:**

Affirmed.

**Key Questions For Authors:**

1. Is there a plan to continuously update this benchmark to cover emerging new software, tasks, and CUA models?
2. Has the benchmark construction and annotation process been automated or semi automated to reduce the cost of future expansion?
3. Would you consider establishing a public leaderboard to continuously track the progress of the community?

**Limitations:**

yes

**Strengths And Weaknesses:**

Strengths:
1. The paper has a clear structure and coherent logic from problem background, benchmark construction, experimental analysis to method proposal.
2. Comprehensive experimental design: The evaluation covers a wide range of visual language models from open source to closed source, from general to specific, from 7B to 235B parameter scales, as well as three representative prompt templates, providing a solid empirical basis for the conclusions.
3. The dataset construction method is rigorous: this paper provides a detailed description of the trajectory collection strategy, manual annotation protocol, and strict quality control process, ensuring the reliability of the benchmark data.

Weakness:
1. The UPE method improved the accuracy and NPV of PRM to 83.1% and 86.2%, while the recall and specificity plummeted to 53.8% and 45.7%, with an overall accuracy of only 50%. The paper acknowledges the decrease in recall/specificity, but lacks sufficient discussion on the serious negative impacts it may cause in scenarios that emphasize "sample efficiency" such as online reinforcement learning, such as slow learning speed and inability to effectively utilize negative samples. This weakens the universality of UPE as a "practical and immediately deployable solution".
2. Table 4 and Table 5 in the paper contain a large number of merged cells and abbreviations, which are too dense and seriously affect readability.

---

> ### Author Rebuttal · Authors · 2026-03-31
>
> **W1: UPE may hurt sample efficiency in RL.**
>
> We agree that the reviewer is identifying a real concern: **UPE is not designed to maximize sample efficiency**, and therefore should not be presented as the right choice for every online RL regime. Its intended role is **reliability-first reward aggregation**, where low-confidence reward assignments may be more harmful than abstention.
>
> To make this trade-off explicit rather than implicit, we added a coverage/abstention analysis. In **PRM**, UPE reaches **83.1% precision** and **86.2% NPV**, but operates at **59.2% coverage** (**40.8% abstention**). Compared to majority voting, UPE improves **NPV by +23.0 [16.2, 29.9]**, while sacrificing efficiency-side metrics. We will revise the paper accordingly and avoid wording that makes UPE sound universally preferable across all RL settings.
>
> At the same time, we would like to clarify that this is **not only an offline discussion** for us. Since submission, we have already started using high-performing CUARewardBench models as reward sources in downstream RL. Concretely, we extended the benchmark with newly released strong single RMs such as **Kimi-K2.5-1T** and **Qwen3.5-397B**, and then used **Qwen3.5-397B** in a preliminary GRPO experiment. This new RL pilot also complements the end-to-end filtering result already reported in Appendix A.6, where reward-model-filtered trajectories improve success rate from **34.68% to 46.10%** on the 47-task Impress benchmark. This is exactly the workflow we envision for CUARewardBench: **benchmark → select reliable RM → deploy in RL**. We therefore view UPE not as a universally optimal solution, but as one reliability-oriented option within a broader, already ongoing direction toward scaling reward-model-based RL for CUA.
>
>
> **W2: Tables 4/5 are too dense.**
>
> Thank you — we agree. The current Tables 4/5 are too dense for comfortable reading, especially with merged cells and many abbreviations. In the revision, we will simplify the table structure, make the abbreviations easier to parse, and move more fine-grained category results to the appendix when appropriate. We appreciate this suggestion and will treat it as a concrete presentation fix rather than leaving it as a vague future improvement.
>
>
> **Q1: Will the benchmark keep evolving?**
>
> Yes — and importantly, this is already underway rather than only planned. After submission, we extended CUARewardBench with newly released strong RMs including **Kimi-K2.5-1T** and **Qwen3.5-397B**. These additions were not merely cosmetic: they immediately changed the set of strongest single-model results and directly informed our downstream RL pilot. In that sense, the benchmark is already functioning as a **living model-selection tool**, not a static one-shot evaluation.
>
> Going forward, we plan to continue extending the benchmark along the same axes that matter most for practical usefulness: broader software coverage, more task types, and stronger/more diverse CUA policies and reward models. We will clarify this continued-update direction in the revision.
>
>
> **Q2: Can benchmark expansion be automated?**
>
> We believe the update cost is manageable, but in different ways for **model-side** and **data-side** expansion.
>
> On the **model side**, the post-submission additions already demonstrate that benchmark maintenance can be lightweight: we evaluated newly released models such as **Kimi-K2.5-1T** and **Qwen3.5-397B** using the same evaluation pipeline, which suggests that keeping CUARewardBench current with new reward models is practically feasible.
>
> On the **data/annotation side**, we see the most promising semi-automation opportunities in **trajectory mining**, **candidate step selection**, and **script-assisted preprocessing**. However, final judgment on **harmful side effects** and **key-step correctness** should remain human-in-the-loop for quality control. So our expected path is not full automation, but **semi-automation with human verification**. We will make this more explicit in the revision.
>
>
> **Q3: Would a public leaderboard be useful?**
>
> Yes — we agree that a public leaderboard would be highly valuable. More importantly, this is already aligned with how we are using CUARewardBench internally. After submission, we used the benchmark to evaluate newly released models such as **Kimi-K2.5-1T** and **Qwen3.5-397B**, and then used the resulting benchmark evidence to select **Qwen3.5-397B** as the reward source for a preliminary downstream RL experiment. In other words, the benchmark is already functioning as an actionable **model-ranking and selection layer**, rather than a paper-only static evaluation.
>
> A public leaderboard would therefore be a natural next step: it would make this workflow visible and reusable for the community, and help the benchmark serve not only as an analysis tool but also as a practical infrastructure layer for reward-model selection in CUA.

---

### Official Review · Reviewer_j7dJ · 2026-03-12

**Soundness:** 2
**Presentation:** 3
**Significance:** 2
**Originality:** 2
**Overall Recommendation:** 4
**Confidence:** 4

**Summary:**

This paper introduces CUARewardBench, a benchmark specifically designed for evaluating reward models on computer-using agent (CUA) trajectories, supporting both trajectory-level (ORM) and step-level (PRM) assessment. Built upon the OSWorld environment, the benchmark comprises 272 trajectory-level annotations and 346 step-level annotations spanning 7 policy models and 10 software categories. Through systematic experiments across 7 vision-language models and 3 prompt templates, the authors reveal limitations of current reward models in visual reasoning and domain knowledge, and find that general-purpose VLMs outperform CUA-specialized models for reward evaluation. Based on these findings, the authors propose Unanimous Prompt Ensemble (UPE), which combines strict unanimous voting with diversified prompt-template configurations to enhance reward model reliability.

**Compliance With Llm Reviewing Policy:**

Affirmed.

**Final Justification:**

Thanks again for the response. The authors re-clarify the contribution, but still, from the perspective of a benchmark paper, the CUARewardBench is built entirely on the single OSWorld environment, making it less convincing. From the perspective of methodology , the novelty is not sufficient. Therefore, I choose to keep my score.

**Key Questions For Authors:**

1.	Statistical reliability: In Table 4, the Writer category contains only 12 success / 14 failure trajectories; in supplementary Table 7, Thunderbird has only 8/8. Given these sample sizes, is it statistically meaningful to report precision and NPV to one decimal place? Could you provide bootstrap confidence intervals, at least for the Overall metrics? If the intervals are wide, this would directly affect my assessment of the benchmark’s reliability.
2.	End-to-end validation of UPE: The experiment in Section A.6 does not use UPE for trajectory filtering. Could you provide a comparative experiment: SFT using training data filtered by UPE (high precision, low recall) versus majority voting (lower precision, higher recall), and compare the resulting agent performance? This is the key experiment for validating the core assumption that “reliability outweighs efficiency.” A positive result would substantially strengthen the paper.
3.	Causal support for findings: Section 3.2 claims that “visual reasoning capability is the core element,” but the difference between GUI-OWL-32B and Qwen2.5VL-32B may stem from multiple factors in the post-training process. Are there controlled experiments that isolate visual reasoning capability as a single variable? If not, should this conclusion be softened to a hypothesis?
4.	Differentiation from concurrent works: Could you select models that are also evaluated in AgentRewardBench and show whether the same model exhibits significantly different rankings across the two benchmarks? This would compellingly demonstrate that CUARewardBench captures reward model capability dimensions unique to desktop operation scenarios.
5.	Annotation agreement: The paper describes a dual-phase annotation workflow but reports no quantitative inter-annotator agreement (e.g., Cohen’s Kappa). Could you provide this metric? What percentage of total annotations were excluded due to failure to reach consensus?

**Limitations:**

Partially addressed. The authors discuss future directions in Section A.8, including PRM challenges, visual reasoning deficiencies, and hybrid verification approaches. However, several key limitations are insufficiently discussed: (1) the benchmark relies entirely on the single OSWorld environment, and generalizability to other desktop environments or real deployment scenarios is not addressed; (2) the impact of the benchmark’s scale on statistical reliability is not acknowledged; (3) the effect of UPE’s high abstention rate on downstream applications is not analyzed. The Impact Statement is overly brief.

**Strengths And Weaknesses:**

Strengths
- The paper targets a genuine and timely research gap. As the CUA community increasingly relies on reward models for both offline trajectory filtering and online RL training, the absence of a systematic evaluation benchmark has been a notable blind spot. This work is the first to simultaneously cover ORM and PRM evaluation for CUA reward models, with a clear problem formulation and practical relevance.
- The annotation methodology is carefully designed, and the annotation quality is well-assured. A particularly notable design choice is the orthogonal annotation strategy: the authors annotate key bad actions within successful trajectories (key_bad_in_success) and key good actions within failed trajectories (key_good_before_bad_in_fail), recognizing that trajectory success and step correctness are independent dimensions. The supplementary materials further reveal fine-grained annotations at critical decision nodes for each trajectory, including post-error recovery step indices, demonstrating substantial annotation depth. The dual-phase annotation workflow (independent annotation followed by cross-validation and consensus discussion, with unresolvable cases excluded) reflects a commitment to quality control.
- Several experimental findings are insightful and informative. For instance, the observation that GUI-OWL-32B, a CUA-specialized model, underperforms its base model Qwen2.5VL-32B suggests that specialized training can be counterproductive when it degrades reasoning capabilities. The complementary precision-NPV trade-offs across different prompt templates provide actionable guidance for model selection and system design in the community.
Weaknesses
- The technical contribution of UPE is extremely limited. Strict unanimous voting is a well-established strategy in classical ensemble learning (Kuncheva, 2014); the only variation here is requiring consensus on both positive and negative predictions—an intuitive design choice rather than a methodological advance. The prompt-template ensemble is purely an empirical combination of existing open-source prompts (ZeroGUI, OpenCUA, SE-WSM), with no theoretical analysis explaining why specific combinations yield complementary effects, nor any automated template selection or weight learning mechanism. Framing this as a “novel ensemble method” and listing it as one of four main contributions constitutes overclaiming. If UPE is removed, the core contribution reduces to the benchmark itself, whose scale (see W2) is insufficient to independently support an ICML paper.
- The benchmark scale is modest, and the statistical reliability of fine-grained analyses is insufficient. While the annotation quality is commendable, the total volume of 272 trajectories (ORM) and 346 step annotations (PRM) limits the robustness of statistical analyses. When split across 10 categories, several categories have very small sample sizes (e.g., Writer with only 12 success / 14 failure trajectories in Table 4; VLC with 12/8 and Thunderbird with 8/8 in supplementary Table 7). At such sample sizes, flipping the judgment on a single sample can shift precision or NPV by several percentage points. The paper reports no confidence intervals, standard deviations, or significance tests throughout. Furthermore, no quantitative inter-annotator agreement metrics (e.g., Cohen’s Kappa) are reported despite the described cross-validation workflow. For a benchmark that positions itself as “comprehensive,” this represents a significant gap in statistical rigor.
- The practical utility of UPE is questionable, and the end-to-end validation is inadequate. UPE improves precision and NPV through strict unanimous voting at the cost of substantially reduced recall and specificity. For instance, in the best ORM UPE configuration, recall drops from 86.3% to 56.8%, meaning nearly half of all successful trajectories are abstained. The authors claim that “reduced sample efficiency can be compensated by increased sampling” but provide no quantitative evidence to support this. The end-to-end experiment in Section A.6 uses only a single domain (Impress) with a single model (Qwen3VL-32B-Thinking) as the reward model for filtering, and critically, does not employ UPE. This experiment therefore cannot validate whether UPE’s high-precision-low-recall filtering strategy actually translates to improved downstream agent performance, leaving the practical value of UPE as an unverified hypothesis.
- Several core findings in Section 3.2 lack causal support and are better characterized as correlational observations. The claim that “visual reasoning capability is the core element of CUA reward models” is primarily based on the observation that GUI-OWL-32B underperforms Qwen2.5VL-32B. However, GUI-OWL’s post-training process involves multiple confounding variables (training data composition, training objectives, number of training steps, etc.), and the authors cannot isolate the effect of reasoning capability degradation. Similarly, the underperformance of Qwen2.5VL-72B relative to the 32B variant is attributed to “additional RL training,” yet the authors themselves hedge with “a possible explanation” before later citing this as a definitive finding. This pattern of escalating tentative observations into firm conclusions recurs throughout the section.
- The differentiation from concurrent works AgentRewardBench (Lù et al., 2025) and Agent-RewardBench (Men et al., 2025) is insufficiently argued. The authors dismiss these works with brief statements (“ignores desktop operations” and “lacks CUA-specific capabilities”), but do not provide cross-benchmark experiments demonstrating that CUARewardBench captures reward model capability dimensions absent from other benchmarks. Given that CUARewardBench is built entirely on the single OSWorld environment, the argument for its unique value and generalizability requires stronger empirical support.
- Section A.9 states “In accordance with ICLR 2026 policy,” but the submission target is ICML 2026, suggesting a residual error from a previous submission. Additionally, the listing of four contributions has a somewhat inflated feel: the second contribution (dataset characteristics) substantially overlaps with the first (benchmark itself). The error analysis appears in near-identical form in both Section A.7 and Section A.10.

---

> ### Author Rebuttal · Authors · 2026-03-31
>
> ## W1
>
> **W1: UPE's technical contribution is limited; strict unanimous voting is well-established.**
>
> We agree UPE is not a new learning algorithm. Its contribution is adapting **abstention-based consensus** to the **two-sided reliability objective** of CUA reward modeling: we optimize for both **precision** and **NPV** by abstaining on disagreements rather than forcing low-confidence labels. Bootstrap analyses confirm stable NPV gains over single RMs with quantified coverage cost. We will revise the wording to present UPE as a **simple but targeted reliability method**, not deep algorithmic novelty.
>
> ## W2 & Q1
>
> **W2: Benchmark scale modest (272 ORM / 346 PRM); no CIs or significance tests. Q1: Could you provide bootstrap CIs?**
>
> Yes. We computed **stratified bootstrap 95% CIs** (2000 iterations, paired resampling):
>
> **Paired delta CIs (UPE vs constituent single RMs):**
>
> | Setting | Comparison | ΔPrecision | ΔNPV |
> |---------|------------|------------|------|
> | ORM | UPE - Q3-32-s | +2.7 [0.1, 5.7] | +12.2 [7.4, 17.0] |
> | ORM | UPE - Q3-235-z | +5.4 [1.7, 9.3] | +4.7 [1.4, 8.6] |
> | PRM | UPE - Q3-32-s | +4.8 [1.0, 9.0] | +19.9 [13.3, 26.8] |
> | PRM | UPE - Q3-235-opencua | +16.5 [10.7, 22.6] | +11.2 [5.6, 17.5] |
>
> All **ΔPrecision** and **ΔNPV** intervals exclude zero. We will keep strongest claims at the **overall** level.
>
> ## W3 & Q2
>
> **W3: UPE's practical utility unvalidated; A.6 does not use UPE. Q2: Could you compare UPE-filtered vs majority-filtered training?**
>
> We agree our end-to-end evidence is **not yet sufficient** for full UPE validation. We provide two narrower pieces of evidence:
>
> **(1) UPE reliability trade-off:**
>
> | Setting | UPE Precision | UPE NPV | Coverage |
> |---------|--------------|---------|----------|
> | ORM | 88.0 [82.9, 93.1] | 95.3 [91.1, 99.0] | 84.9% |
> | PRM | 83.1 [76.9, 89.2] | 86.2 [79.3, 93.1] | 59.2% |
>
> Compared to majority voting, UPE gains **+15.0 [9.7, 20.0] ΔNPV** on ORM and **+23.0 [16.2, 29.9]** on PRM, because majority voting forces disagreements into negatives while UPE abstains.
>
> **(2) Preliminary RL:** We performed **GRPO on Qwen3VL-4B-Thinking** in the Impress domain from OSWorld (32-task feasible subset, SR range 0.12–1.0), comparing OSWorld predefined evaluator rewards vs. **Qwen3.5-397B** RM rewards:
>
> | Steps | Evaluator SR | RM SR |
> |-------|------------|-------|
> | 1–10 | 0.557 | 0.540 |
> | 11–20 | 0.649 | 0.578 |
> | 21–30 | 0.705 | 0.627 |
>
> This complements Appendix A.6 filtering (SR: 34.68%→46.10%). We regard UPE-in-the-loop RL as an important next step and are actively pursuing it.
>
> ## W4 & Q3
>
> **W4: Section 3.2 findings lack causal support. Q3: Can you isolate visual reasoning?**
>
> We agree. Current evidence is **correlational**, not a controlled causal isolation. We will revise **"is the core element"** to **"appears to be a dominant factor"** / **"our results suggest."**
>
> ## W5 & Q4
>
> **W5: Differentiation from concurrent benchmarks insufficient. Q4: Cross-benchmark ranking comparison?**
>
> CUARewardBench is **complementary**: it targets **desktop computer-use** across 10 software categories with **ORM + PRM** key-step annotations, emphasizing **NPV/specificity** for negative-side reliability. Concurrent benchmarks focus on web-agent or broader multimodal settings. We did not complete a cross-benchmark ranking analysis but will add clearer positioning and treat this as an important future extension.
>
> ## W6
>
> **W6: Residual "ICLR 2026 policy" error; inflated claims; redundant appendix.**
>
> We will correct the residual wording, tighten contribution statements, and remove redundancy between A.7 and A.10.
>
> ## Q5
>
> **Q5: Inter-annotator agreement and exclusion rate?**
>
> CUARewardBench labels are **not raw single-annotator labels**. Our **dual-phase consensus protocol** (Appendix A.2): (1) independent annotation with justifications, (2) cross-validation and joint re-examination; cases without consensus are **excluded**. For PRM, we annotate only **key good/bad actions** and exclude ambiguous tasks. The benchmark is a **consensus-only dataset**, prioritizing label quality over size.

---

> > ### Author Rebuttal · Reviewer_j7dJ · 2026-04-03
> >
> > Thank the authors for the detailed rebuttal. Part of my concerns are addressed, but the technical contribution is not convincing enough.

---

> > > ### Author Response · Authors · 2026-04-04
> > >
> > > Thank you for the follow-up. We agree that **UPE itself is not a deep new learning algorithm**, and we do not want the paper to be judged as if its core claim were algorithmic novelty alone.
> > >
> > > In other words, we believe the paper should be judged less as a standalone algorithm paper and more as a **benchmark-and-methodology paper for CUA reward modeling**. The technical contribution we aim to make is the following combination:
> > > (1) **CUARewardBench**, to our knowledge, is the first benchmark centered on **desktop computer-use reward modeling**, rather than web-only or broader multimodal-agent settings;
> > > (2) it evaluates **both ORM and PRM**, with a **consensus-only annotation protocol** and an explicit focus on **two-sided reward reliability** (including NPV/specificity, not only precision);
> > > (3) it produces actionable empirical findings about reward-model behavior in desktop CUA settings; and
> > > (4) **UPE** is a deliberately simple but targeted reliability-oriented method derived from these findings, using abstention-based consensus to improve reward reliability, with the trade-off now made explicit through our added bootstrap and coverage analyses.
> > >
> > > We therefore agree that the paper should not be read as claiming deep algorithmic novelty from UPE alone. Rather, its contribution is a **benchmark + empirical analysis + practically motivated reliability method** package for CUA reward modeling. This is also why we added downstream evidence: Appendix A.6 already shows that reward-model-based filtering improves task success rate (**34.68% → 46.10%**), and our new RL pilot shows that a high-performing benchmark RM can already serve as a useful reward source in online training. We will further revise the wording to make this scope and positioning as explicit as possible.

---

### Official Review · Reviewer_Zya1 · 2026-03-12

**Soundness:** 3
**Presentation:** 2
**Significance:** 2
**Originality:** 2
**Overall Recommendation:** 4
**Confidence:** 4

**Summary:**

This paper introduces CUARewardBench, the first comprehensive benchmark for evaluating both outcome reward models (ORM) and process reward models (PRM) on computer-using agent (CUA) tasks. The benchmark encompasses trajectories from 10 software categories and 7 agent architectures with varying performance levels (25.9%-50.8% success rates), all expertly annotated through carefully designed protocols with rigorous quality control. Through extensive experiments across 7 vision-language models and 3 prompt templates, the authors reveal critical limitations of current CUA reward models, including insufficient visual reasoning capabilities, knowledge deficiencies, and the counterintuitive finding that general VLMs outperform specialized CUA models for reward evaluation. Based on these insights, the authors propose Unanimous Prompt Ensemble (UPE), a novel ensemble method that achieves 88.0% precision and 95.3% NPV for ORM, and 83.1% precision and 86.2% NPV for PRM, substantially outperforming single VLMs and traditional ensemble approaches. The work addresses an important gap in the CUA evaluation landscape where script-based verifiers suffer from limited scalability and inability to provide step-wise assessment.

**Compliance With Llm Reviewing Policy:**

Affirmed.

**Final Justification:**

The authors rebuttal has fully resolved my concerns with solid experimental results, so I am raising my score to 4.

**Key Questions For Authors:**

1. How does CUARewardBench compare to the concurrent AgentRewardBench (Liu et al., 2025) in terms of coverage, annotation quality, and evaluation insights? A discussion of the relationship and complementary aspects would be valuable.

2. What is the inter-annotator agreement for both ORM and PRM annotations? This is critical for establishing the benchmark's reliability, especially for PRM where step-level correctness can be subjective.

3. Has the UPE method been tested for filtering training data in an actual RL training loop for CUA agents? Demonstrating downstream utility (e.g., improved agent performance when using UPE-filtered trajectories for training) would significantly strengthen the practical contribution.

4. Why do general VLMs outperform specialized CUA models for reward evaluation? Can the authors provide a deeper analysis of the failure model: is it due to the specialized models' narrow training distribution, or fundamental architectural differences?

5. How sensitive is the UPE method to the specific choice of prompt templates? The paper uses 3 templates, but the unanimity requirement means the choice of templates critically affects both precision and recall.

**Limitations:**

Yes.

**Strengths And Weaknesses:**

Strengths:

1. The paper addresses a relevant problem: evaluating reward models for computer use agent trajectories, which is becoming increasingly important as RL-based training of CUA agents gains traction.

2. The UPE (Unanimous Precision Evaluation) metric is a reasonable proposal for handling the challenge of evaluating step-level rewards, though it is conceptually simple.

Weaknesses:

1. The UPE metric has significant limitations: While unanimous voting increases precision, it dramatically reduces recall—any step where annotators disagree is excluded. This means the benchmark may systematically ignore the most interesting and informative edge cases where reward model evaluation is most needed. The paper does not adequately quantify how much data is lost through unanimous filtering or analyze whether the retained samples are representative of the full difficulty distribution.

2. Counterintuitive finding undermines benchmark utility: The paper reports that general-purpose VLMs outperform specialized CUA models on reward evaluation. Rather than presenting this as an interesting finding, this should raise concerns about what the benchmark is actually measuring. If specialized models trained on CUA trajectories perform worse than general models, it suggests either (a) the benchmark tasks don't require CUA-specific knowledge, (b) the evaluation protocol is flawed, or (c) the specialized models are poorly trained. The paper does not adequately investigate which explanation holds.

3. No downstream validation: The benchmark evaluates reward models in isolation but provides no evidence that benchmark performance correlates with actual downstream utility—e.g., using the evaluated reward models for RL training of CUA agents and measuring agent performance. Without this validation, the benchmark's practical value remains undemonstrated. A reward model benchmark that doesn't predict downstream training outcomes is of limited scientific value.

4. Limited scale and diversity of trajectories: The benchmark likely contains a relatively small number of annotated trajectories. For a benchmark paper, the scale should be sufficient to enable statistically significant comparisons across many models. The paper does not report confidence intervals or statistical significance tests for the reported results.

5. Missing comparison to simpler baselines: The paper does not compare against simple heuristic reward functions (e.g., screenshot similarity, task completion detection via template matching) which could serve as important baselines for understanding the difficulty of the benchmark tasks.

---

> ### Author Rebuttal · Authors · 2026-03-31
>
> ## W1 & Q3
>
> **W1: UPE may ignore informative edge cases; data loss not quantified. Q3: Has UPE been tested in an actual RL loop?**
>
> UPE is designed for **high-confidence reward signals**, not maximum coverage. We quantified its abstention:
>
> **Overall:** ORM UPE retains **84.9%** (Pos 82.7%, Neg 87.2%); PRM retains **59.2%** (Pos 60.4%, Neg 57.9%).
>
> **Per-category (ORM):** coverage ranges **75.0%–88.9%** (std=4.7) across all 10 categories—no category collapses. At the task level, **85/122 tasks (69.7%) have zero abstentions**; max is 2. Disagreement is spread thinly, not concentrated.
>
> Regarding downstream RL: we do **not** claim full UPE-in-the-loop validation yet. We performed **GRPO on Qwen3VL-4B-Thinking** in the Impress domain from OSWorld (32-task feasible subset, SR range 0.12–1.0), comparing OSWorld predefined evaluator rewards vs. **Qwen3.5-397B** RM rewards:
>
> | Steps | Evaluator SR | RM SR |
> |-------|------------|-------|
> | 1–10 | 0.557 | 0.540 |
> | 11–20 | 0.649 | 0.578 |
> | 21–30 | 0.705 | 0.627 |
>
> This complements the filtering result in Appendix A.6 (SR: 34.68%→46.10%). We regard UPE-in-the-loop RL as an important next step and are actively pursuing it.
>
> ## W2 & Q4
>
> **W2: General VLMs outperforming specialized CUA models undermines benchmark utility. Q4: What explains this?**
>
> We agree the original framing is too coarse. CUARewardBench evaluates **reward modeling** (judging correctness), not **policy execution**. Specialized CUA post-training may improve action execution while not preserving visual reasoning needed for reward evaluation—these are distinct capabilities. We will rewrite the "general > specialized" framing, soften causal language to **"our results suggest"**, and present this as a **supported hypothesis**, not a causal conclusion.
>
> ## W3
>
> **W3: No downstream validation; benchmark's practical value undemonstrated.**
>
> Post-submission, we evaluated newer models. **Qwen3.5-397B** achieves the highest precision (**89.3**) and specificity (**89.5**) among all single RMs; we also added **Kimi-K2.5-1T** (P=88.4, OA=90.1). We used Qwen3.5-397B in the GRPO experiment (W1&Q3). Combined with Appendix A.6 filtering (SR: 34.68%→46.10%), this provides **two complementary downstream signals**—trajectory filtering and online RL—both using benchmark-identified strong RMs.
>
> ## W4
>
> **W4: Limited scale; no CIs or significance tests.**
>
> We computed **stratified bootstrap 95% CIs** (2000 iterations, paired resampling, 272 ORM / 346 PRM samples):
>
> **Paired delta CIs (UPE vs constituent single RMs):**
>
> | Setting | Comparison | ΔPrecision | ΔNPV |
> |---------|------------|------------|------|
> | ORM | UPE - Q3-32-s | +2.7 [0.1, 5.7] | +12.2 [7.4, 17.0] |
> | ORM | UPE - Q3-235-z | +5.4 [1.7, 9.3] | +4.7 [1.4, 8.6] |
> | PRM | UPE - Q3-32-s | +4.8 [1.0, 9.0] | +19.9 [13.3, 26.8] |
> | PRM | UPE - Q3-235-opencua | +16.5 [10.7, 22.6] | +11.2 [5.6, 17.5] |
>
> All **ΔPrecision** and **ΔNPV** intervals exclude zero. We will limit strongest claims to the **overall** level.
>
> ## W5
>
> **W5: Missing heuristic baselines (screenshot similarity, template matching).**
>
> We respectfully disagree that these are well-aligned baselines. (1) Our paper exists precisely because verifier-style approaches are **not scalable** and lack step-wise assessment. (2) Screenshot similarity requires task-specific target states, reverting to handcrafted verification; template matching requires per-task rules, falling into the same non-scalable regime. (3) CUARewardBench covers **both ORM and PRM** with step-level annotations—beyond what end-state heuristics can capture. We discuss **hybrid verification** as a future direction.
>
> ## Q1
>
> **Q1: How does CUARewardBench compare to concurrent AgentRewardBench?**
>
> CUARewardBench is **complementary**: it targets **desktop computer-use** across 10 software categories with both **ORM and PRM** key-step annotations, emphasizing **NPV/specificity** for negative-side reliability. Concurrent benchmarks focus on web-agent or broader multimodal-agent settings. Our claim is not that they are insufficient, but that they leave a **specific gap** in desktop CUA reward evaluation.
>
> ## Q2
>
> **Q2: Inter-annotator agreement and annotation reliability.**
>
> CUARewardBench labels are **not raw single-annotator labels**. Our **dual-phase consensus protocol** (Appendix A.2): (1) independent annotation with justifications, (2) cross-validation and joint re-examination; cases without consensus are **excluded**. For PRM, we annotate only **key good/bad actions** and exclude ambiguous tasks. The benchmark is a **consensus-only dataset**.
>
> ## Q5
>
> **Q5: How sensitive is UPE to prompt template choice?**
>
> UPE benefits from **template diversity** rather than any single privileged template. The paper analyzes three open-sourced templates showing complementary trade-offs. We will moderate the claim: UPE leverages **complementary prompt behaviors**, not a uniquely chosen template.

---

> > ### Author Rebuttal · Reviewer_Zya1 · 2026-04-04
> >
> > Thanks for the authors' solid rebuttal. I will raise my score to 4.

---

> > > ### Author Response · Authors · 2026-04-04
> > >
> > > Thank you very much for your thoughtful follow-up and for your positive assessment of our rebuttal. We sincerely appreciate your time and consideration, and we are glad that our clarifications helped address your concerns.

---

### Official Review · Reviewer_nP7F · 2026-03-13

**Soundness:** 3
**Presentation:** 3
**Significance:** 3
**Originality:** 3
**Overall Recommendation:** 4
**Confidence:** 2

**Summary:**

This paper studies reward modeling for computer-using agents (CUAs), which interact with graphical user interfaces to accomplish complex tasks. The authors introduce CUARewardBench, a benchmark for evaluating reward models in computer-using environments. The benchmark supports evaluation at two levels, both the outcome reward modeling (ORM) which predicts whether an entire trajectory successfully completes a task, and process reward modeling (PRM) which evaluates whether individual agent actions are correct or helpful toward task completion.

The dataset covers 10 software categories and 7 agent architectures with varying performance levels, providing diverse task difficulty and agent behaviors. The authors evaluate several vision-language models and find that strong visual reasoning and prompt design significantly affect reward prediction performance. They also propose Unanimous Prompt Ensemble (UPE), which combines multiple models and prompts using unanimous voting to improve reward reliability.

**Compliance With Llm Reviewing Policy:**

Affirmed.

**Ethical Review Flag:**

Flag this paper for an ethics review.

**Key Questions For Authors:**

1. Strick unanimous voting decreases the recall by quite a lot for the 3-model ensemble, e.g. in Table 4/5 by quite a lot, to what extent is this acceptable?
2. Given that the benchmarks contain <300 trajectories, how statistically significant is the performance difference between these models being studied?

**Limitations:**

Yes

**Strengths And Weaknesses:**

Strengths: The annotator workflow is clearly described. The construction pipeline excluded ambiguous tasks, balanced the dataset across software categories, and explicitly annotated good actions / bad actions within failed or successful trajectories. For experiement analysis, different VLMs (e.g., general reasoning model and specialized CUA models) and multiple exsiting prompt templates are tested. The end-to-end experiment seems rigorous.

Weakness: the strick unanimous voting (3.4) seem to specifically decrease recall by quite a lot according to the empircal results. 27 trajectories per software domain seems like a small sample per domain.

---

> ### Author Rebuttal · Authors · 2026-03-31
>
> ## W1 & Q1
>
> **W1: Strict unanimous voting substantially decreases recall. Q1: To what extent is this acceptable?**
>
> We agree that UPE introduces an explicit reliability-efficiency trade-off. In reward filtering and RL reward provision, **incorrect reward signals are often more harmful than reduced sample efficiency**, so we prioritize precision/NPV over recall/specificity. While the reviewer asks about the 3-model ensemble in Tables 4/5, we report the detailed coverage analysis on the updated 2-member setting, which quantifies the same mechanism: UPE abstains on disagreement rather than forcing low-confidence labels.
>
> **ORM UPE Coverage (2-member: Q3-32-s + Q3-235-z):** 84.9% overall (Pos 82.7%, Neg 87.2%), abstaining on **15.1%** of samples.
>
> **Per-category ORM coverage (UPE):**
>
> | Category | Total | Abstain | Cov% |
> |----------|-------|---------|------|
> | chrome | 30 | 4 | 86.7 |
> | gimp | 26 | 5 | 80.8 |
> | calc | 34 | 4 | 88.2 |
> | impress | 28 | 6 | 78.6 |
> | writer | 26 | 3 | 88.5 |
> | multi_apps | 36 | 4 | 88.9 |
> | os | 26 | 3 | 88.5 |
> | thunderbird | 16 | 3 | 81.2 |
> | vlc | 20 | 5 | 75.0 |
> | vs_code | 30 | 4 | 86.7 |
>
> Coverage ranges **75.0%–88.9%** (std=4.7) with no category collapse. At the task level, **85/122 tasks (69.7%) have zero abstentions**; max is 2. Disagreements are spread thinly rather than concentrated.
>
> In exchange, UPE gains **+15.0 [9.7, 20.0] ΔNPV** over majority voting (bootstrap interval excludes zero). We will present UPE as a **reliability-oriented aggregation strategy**, not a one-size-fits-all solution.
>
> ## W2 & Q2
>
> **W2: Benchmark scale is modest (~27 per domain); differences may not be significant. Q2: Can you provide statistical evidence?**
>
> Yes. All methods are evaluated on the same **272 ORM trajectories** and **346 PRM annotated steps**, so we used **stratified bootstrap 95% CIs with paired resampling** (2000 iterations):
>
> **ORM Bootstrap 95% CI (2-member: Q3-32-s + Q3-235-z):**
>
> | Method | Precision | NPV | Recall | Specificity |
> |--------|-----------|-----|--------|-------------|
> | Q3-32-s | 85.3 [80.1, 90.4] | 83.1 [78.1, 88.5] | 83.5 [77.7, 89.2] | 85.0 [78.9, 91.0] |
> | Q3-235-z | 82.6 [77.7, 87.6] | 90.6 [85.4, 95.5] | 92.1 [87.1, 96.4] | 79.7 [72.9, 86.5] |
> | Majority | 88.0 [82.9, 93.1] | 80.3 [75.3, 85.7] | 79.1 [71.9, 85.6] | 88.7 [83.5, 94.0] |
> | UPE | 88.0 [82.9, 93.1] | 95.3 [91.1, 99.0] | 79.1 [71.9, 85.6] | 75.9 [68.4, 82.7] |
>
> **Paired delta CIs (UPE vs constituent single RMs):**
>
> | Setting | Comparison | ΔPrecision | ΔNPV |
> |---------|------------|------------|------|
> | ORM | UPE - Q3-32-s | +2.7 [0.1, 5.7] | +12.2 [7.4, 17.0] |
> | ORM | UPE - Q3-235-z | +5.4 [1.7, 9.3] | +4.7 [1.4, 8.6] |
> | PRM | UPE - Q3-32-s | +4.8 [1.0, 9.0] | +19.9 [13.3, 26.8] |
> | PRM | UPE - Q3-235-opencua | +16.5 [10.7, 22.6] | +11.2 [5.6, 17.5] |
>
> All **ΔPrecision** and **ΔNPV** intervals exclude zero. We will limit our strongest claims to the **overall** level and avoid over-interpreting per-category differences.

---

> > ### Author Rebuttal · Reviewer_nP7F · 2026-04-05
> >
> > Thank you for the rebuttal: my questions have largely been addressed, and I will maintain my positive score.

---

> > > ### Author Response · Authors · 2026-04-06
> > >
> > > Thank you very much for your thoughtful follow-up and positive acknowledgement. We are glad that our rebuttal helped clarify the main concerns, and we sincerely appreciate your time and constructive feedback throughout the review process.

---

### Decision · Program_Chairs · 2026-04-30

**Decision:**

Accept (regular)

**Comment:**

This paper studies reward modeling for computer-using agents (CUAs), which interact with graphical user interfaces to accomplish complex tasks. The authors introduce CUARewardBench, a benchmark designed to evaluate reward models on CUA trajectories at two levels: outcome reward modeling (ORM), which predicts whether an entire trajectory successfully completes a task, and process reward modeling (PRM), which evaluates whether individual actions are correct or helpful toward task completion. The benchmark covers 10 software categories and 7 agent architectures with varying performance levels, and relies on expert annotation protocols with careful quality control.

All reviewers agreed that the paper addresses a relevant and timely problem, given the growing importance of evaluating reward models for CUA trajectories and the increasing use of RL-based training for computer-using agents. Reviewers also appreciated that the work helps fill an important gap in the current evaluation landscape, where script-based verifiers can be difficult to scale and do not naturally support step-level assessment. They further highlighted the care taken in the annotation pipeline, including the exclusion of ambiguous tasks, balancing across software categories, and explicit annotation of good and bad actions within successful and failed trajectories. Finally, reviewers discussed UPE, the proposed Unanimous Prompt Ensemble method, which uses strict unanimous voting across prompt configurations to improve reward-model reliability.

At the same time, reviewers raised several important concerns. One reviewer raised concerns about the limitations of UPE, the lack of downstream validation, the modest scale of the benchmark, and the absence of simpler heuristic baselines. The authors provided a thorough response to these points, and the reviewer indicated that their concerns had been addressed, raising their score. Similar concerns about UPE’s trade-offs and the benchmark’s scale were also raised by other reviewers. In particular, one reviewer remained unconvinced about the significance of the technical contribution associated with UPE, arguing that the core idea of unanimous voting is well established and that the paper would benefit from a stronger justification for why this adaptation should be viewed as a meaningful methodological advance. Although the authors’ rebuttal addressed some of this reviewer’s other concerns, including by adding bootstrap confidence intervals and clarifying the scope of their claims, the reviewer still felt that the methodological novelty was not sufficiently convincing. Another reviewer expressed a similar concern about the recall and sample-efficiency tradeoff of UPE, noting that this weakens the universality of the method as "*a practical and immediately deployable solution*" (as claimed in the paper). Finally, the paper’s differentiation from concurrent benchmarks such as AgentRewardBench and Agent-RewardBench was also questioned. The authors clarified that CUARewardBench is intended to be complementary and emphasized its focus on desktop computer use across multiple software categories with both ORM and PRM annotations, but one reviewer felt that the argument for its unique value and broader generalizability could still be strengthened, especially given that the benchmark is built entirely on the OSWorld environment.

Overall, reviewers agreed that this paper studies an important problem and that building reliable reward-model benchmarks for computer-using agents is likely to become increasingly valuable. They appreciated the care taken in the benchmark construction and the effort to support both trajectory-level and step-level evaluation, while also identifying questions about scale, validation, and the significance of some of the technical claims. They encouraged the authors to continue strengthening the manuscript based on the points raised in the reviews, particularly by clarifying the benchmark’s positioning relative to concurrent work and by further substantiating the practical value of the proposed evaluation framework. Addressing these points would further strengthen the paper and help highlight its contributions more clearly.